# Mechanically robust and personalized silk fibroin-magnesium composite scaffolds with water-responsive shape-memory for irregular bone regeneration

Zhinan Mao[1,2,6], Xuewei Bi[1,2,6], Chunhao Yu[2], Lei Chen[3], Jie Shen[1], Yongcan Huang[1], Zihong Wu[4], Hui Qi[3], Juan Guan [5], Xiong Shu[3] ✉, Binsheng Yu[1] ✉ & Yufeng Zheng [2] ✉

The regeneration of critical-size bone defects, especially those with irregular shapes, remains a clinical challenge. Various biomaterials have been developed to enhance bone regeneration, but the limitations on the shape-adaptive capacity, the complexity of clinical operation, and the unsatisfied osteogenic bioactivity have greatly restricted their clinical application. In this work, we construct a mechanically robust, tailorable and water-responsive shape-memory silk fibroin/magnesium (SF/MgO) composite scaffold, which is able to quickly match irregular defects by simple trimming, thus leading to good interface integration. We demonstrate that the SF/MgO scaffold exhibits excellent mechanical stability and structure retention during the degradative process with the potential for supporting ability in defective areas. This scaffold further promotes the proliferation, adhesion and migration of osteoblasts and the osteogenic differentiation of bone marrow mesenchymal stem cells (BMSCs) in vitro. With suitable MgO content, the scaffold exhibits good histocompatibility, low foreign-body reactions (FBRs), significant ectopic mineralisation and angiogenesis. Skull defect experiments on male rats demonstrate that the cell-free SF/MgO scaffold markedly enhances bone regeneration of cranial defects. Taken together, the mechanically robust, personalised and bioactive scaffold with water-responsive shape-memory may be a promising biomaterial for clinical-size and irregular bone defect regeneration.

Repair of critical-size irregular bone defects caused by trauma, tumour or infection is a major challenge for orthopaedic surgeons and basic science scientists[1,2]. Autologous bone grafting is still the golden standard for treating these bone defects[3]; nevertheless, autografts are frequently associated with limitations such as wound problems, donor site infection, pain, sensory loss and even reoperation[4,5]. In clinics, the alternatives to autologous or allogenic bone grafts, such as titanium alloy and polyether ether ketone, have excellent mechanical properties[6,7], but their inherent bioinertness results in poor integration between the grafts and host bone tissue, leading to loosening, detachment and ineffective functional bone repair[8–10]. Importantly, most existing bone substitutes lack shape adaptability, leading to

**Fig. 1 | Design and working mechanisms of the SF/MgO composite scaffolds for minimally invasive clinical-size irregular bone regeneration. a** Preparation process of the SF/MgO composite scaffolds and its mechanism of cross-linking reaction: the epoxy group of EGDE reacts with the phenyl hydroxyl and amino groups from SF, and the SF molecules form a physically cross-linked network via the β-sheet domains; electrostatic interactions occur between the negatively charged SF and positively charged MgO particles; according to the mass ratio of the nano-MgO particles to the solute mass of the SF solution at 0 wt%, 10 wt% and 30 wt %, named SF, SF-1nMgO and SF-3nMgO, respectively. **b** The SF/MgO composite scaffolds are tailored to the actual needs of clinical surgery; after being implanted, the cell-free SF/MgO composite scaffolds are recovered by a water-responsive shape-memory effect, resulting in tight contact with the surrounding tissue; the $Mg^{2+}$ is released slowly from the SF/MgO scaffolds to regulate the functions of migrating cells, including adhesion, proliferation and vascularisation, and to promote the expression of related osteogenic genes (runt-related transcription factor 2 (Runx2), osteocalcin (OCN), osteopontin (OPN) and collagen I (COL I)); finally, the defective bone is repaired in situ.

insufficient matching for irregular defects[11,12]. In open surgery, irregular defects are usually enlarged into regular ones; however, these procedures increase the difficulty and cost of surgery and bring severe trauma, pain and dysfunction to patients. Therefore, it is worth developing a structurally stable and shape-memory porous scaffold that can adapt to the irregular-shaped bone defect to achieve minimally invasive treatment.

Bone substitutes have gained considerable attention to meet the demands of enhanced recovery after surgery (ERAS) by minimising surgical trauma[13]. Shape-memory polymers (SMPs), a kind of emerging intelligent biomaterial, have brought great hope to patients with bone defects[14–18]. SMPs can transform between initial shapes and programmed shapes under various stimuli, such as light[19,20], heat[21], electrical and magnetic fields[22] and water[23]. SMPs can be implanted using minimally invasive methods and matched the irregular bone defect shape, thus strengthening the tightness of the interface between the materials and the surrounding tissue[19,20,24]. For biomedical implants, the most physiologically accessible shape-recovery stimuli are temperature and water. Therefore, SMPs can be recovered the original shape by water/blood, it will become a highly attractive approach to meet the ERAS requirements and benefit patients, and inevitably decrease the complexity, cost and time of operations, especially when treating irregular bone defects.

Silk fibroin (SF), derived from the cocoons of silkworms, is a natural protein biomaterial with a number of merits, such as easy accessibility, good biocompatibility, controllable biodegradation and excellent mechanical properties, making it suitable for wide application in the field of tissue regeneration, drug delivery and medical implants, in the form of hydrogels and scaffolds[25–31]. However, traditional SF materials do not have a shape-memory effect, and their structure is permanently damaged under external force, resulting in a poor integration fit with bone defects after implantation. Meanwhile, the implantable shape-memory SF materials must also be sufficiently tough and fatigue-resistant, and exhibit a recovery force strong enough to expand in a confined space. More importantly, bone repair is a complex process that involves many types of cells, signalling molecules and multiple growth factors. All these biochemical cues can promote bone formation, accelerate bone healing and improve the quality of bone repair by triggering cell proliferation and differentiation. To improve the biological activity of SF materials, they are generally used in conjunction with other bioactive factors or cells[32,33]. However, biomaterials containing cells and growth factors are clinically limited due to ethical and biosafety concerns[34].

Magnesium (Mg) is an essential element for bone health and diseases because it is involved in the physiological processes of bone tissue formation, bone metabolism and bone mineral crystallisation[35,36]. In humans, approximately 60% of Mg is stored in the bone matrix[37]. Studies have proved that Mg could regulate cell functions, including proliferation, adhesion and migration, and promote neovascularisation[38,39]. The viability and osteogenic differentiation properties of pre-osteoblasts and mesenchymal stem cells (MSCs) could also be upregulated by magnesium ion ($Mg^{2+}$) at 50 to 100 ppm[38,40]. However, the rapid degradation and subsequent hydrogen release of Mg and its alloys in vivo is an urgent problem for researchers. By contrast, magnesium oxide (MgO), an oxide of Mg, neither degrades rapidly nor produces hydrogen in vivo[41], and it could thus be used as a source of $Mg^{2+}$ in bone tissue materials. Therefore, we introduce MgO particles to shape-memory SF scaffolds to regulate their osteoinductive function and retain their shape-memory ability. Meanwhile, as SF is negatively charged and MgO particles are positively charged, electrostatic interactions can occur between them, which further promotes the formation of multiple cross-linked structures in the SF system.

In this work, we prepare a mechanically robust, water-responsive and personalised SF/MgO shape-memory porous scaffold. The characteristics of the scaffold, including excellent mechanical stability and structure retention, excellent biological performances and a water-responsive shape-memory effect that enables it to tight contact with the surrounding tissues. After the implantation of SF/MgO scaffolds, $Mg^{2+}$ is released slowly with the degradation of MgO particles to regulate the functions of migrating cells, and to promote the expression of related osteogenic genes; finally, the critical-size defective bone is repaired in situ (Fig. 1b).

## Results

### Design and characterisation of bioactive silk fibroin/magnesium composite scaffolds

To fabricate structurally stable, shape-memory and personalised porous scaffolds for irregular bone defects, the SF/MgO composite scaffolds were fabricated (Fig. 1a, b). Detailed experimental details are described in Supplementary Information and Supplementary Fig. 1. The SF solution (60 mg/ml) was mixed with aqueous ethylene glycol diglycidyl ether (EGDE, 2 mmol/g) and N,N,N′,N′-Tetramethylethylenediamine (TEMED, 0.25 v/v%). Then, nano-MgO particles with different contents (0 wt%, 10 wt%, and 30 wt%, relative to the weight of the SF solute) were added to the SF solution, and the SF/MgO (SF, SF-1nMgO, SF-3nMgO) mixture solution was prepared (Fig. 1a and Supplementary Fig. 1). After 24 h of cryogelation (−10 °C), the SF/MgO composite scaffolds were constructed. As shown in Fig. 2a and Supplementary Figs. 2, 3, the changes before and after the mixed solution gel could be clearly seen. During the process, the phenyl hydroxyl and amino groups of the SF chains could react with the epoxy group of EGDE, and the SF chains formed a β-sheet crystalline structure through self-assembly as physical crosslinks. The positive surface charge of the MgO particles enables loading with negatively charged SF molecules by a simple charge-charge interaction. Scanning electron microscopy (SEM) was used to observe the surface morphology and MgO particle distribution. The SF/MgO composite scaffolds had uniform porous microstructures with ~100 μm size pores (Fig. 2a and Supplementary Figs. 2, 3). Moreover, energy-dispersive spectrometry maps confirmed that MgO particles were homogeneously distributed on the surface and inside the pore wall of the SF/MgO scaffolds (Fig. 2a and Supplementary Figs. 2, 3).

The chemical compositions and secondary structure content of pure SF scaffolds and the SF/MgO composite scaffolds were analysed using Fourier-transform infra-red spectroscopy (FTIR) and X-ray diffraction (XRD) (Fig. 2b, d). As shown in Fig. 2b, the absorption bands showed around 3700 $cm^{-1}$ corresponding to the characteristic peaks of MgO particles, which could be observed in SF/MgO scaffolds. The content of β-sheet in SF scaffolds is critical for their mechanical strength and degradation behaviour. The characteristic vibrational bands of SF at 1700−1450 $cm^{-1}$ for amide I (C = O stretching) and amide II (secondary NH bending) can be observed by FTIR. In the amide I band, random coils and α-helix structures were generally found at 1640−1660 $cm^{-1}$, whereas the β-sheet structure was commonly identified at 1620−1637 $cm^{-1}$ [42]. Notably, the strong absorption peak of SF scaffolds and SF/MgO composite scaffolds at 1621 $cm^{-1}$ indicated that their secondary structure was dominated by the β-sheet conformation. To further analyse the secondary structure content, the amide I region of the FTIR spectra was analysed using Fourier self-deconvolution (Supplementary Fig. 4). As shown in Fig. 2c, there was no statistical difference in the β-sheet contents among the three scaffolds. The results indicated that the addition of MgO particles could not further promote the crystallisation of SF scaffolds. The main reason for our analysis findings may be that the chemical cross-linking in cryogelation promoted the formation of more β-sheet conformations in the SF scaffolds[30]. XRD analysis was used to further verify the existence of MgO particles in the scaffolds (Fig. 2d). Due to the existence of MgO particles, the typical peaks appear at 2θ = 18.5° and 42.8° in the corresponding XRD patterns of the SF/MgO composite scaffolds, while no

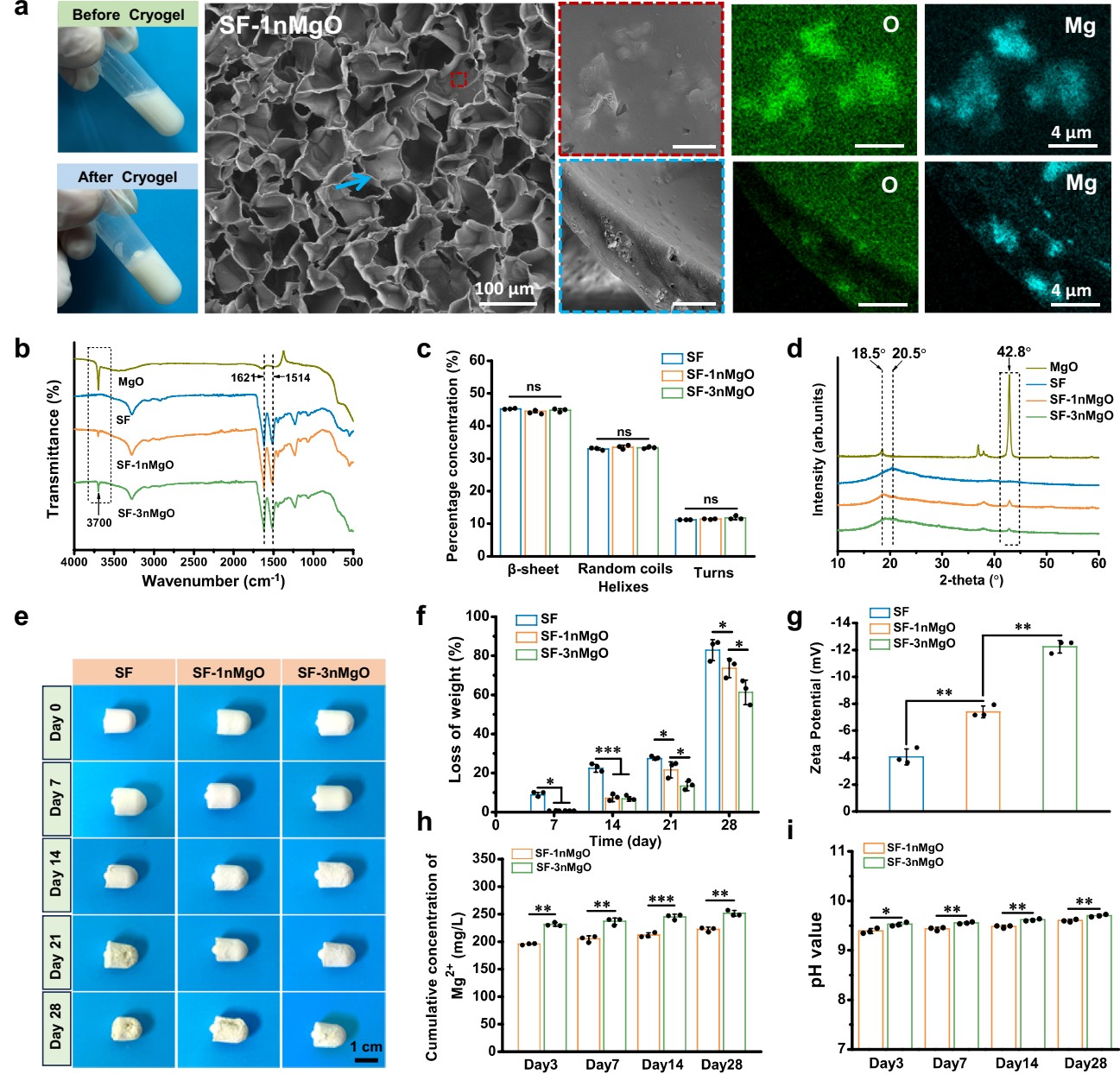

**Fig. 2 | Morphology and characterisation of SF and SF/MgO composite scaffolds. a** Macro-morphology and SEM image of SF-1nMgO composite scaffolds; energy-dispersive spectrometry maps and spectra of the SF-1nMgO composite scaffolds; red frame and blue arrows indicate the surface and the pore wall of the scaffolds; green and blue correspond to elemental oxygen (O) and Mg, respectively. **b** FTIR spectra of the MgO, SF and SF/MgO scaffolds. **c** Conformation contents of SF and SF/MgO scaffolds from peak deconvolution analysis of the amide I region. **d** XRD patterns of the MgO, SF and SF/MgO scaffolds. **e** Macroscopic appearance of SF, SF-1nMgO and SF-3nMgO scaffolds after degradation in protease XIV solutions at different time points (7, 14, 21, 28 d). **f** Quantitative statistics of degradation rates of SF, SF-1nMgO and SF-3nMgO scaffolds. **g** Zeta potential of SF, SF-1nMgO and SF-3nMgO solutions. **h** Accumulative $Mg^{2+}$ release from SF-1nMgO and SF-3nMgO scaffolds immersed in SBF for different time lengths. **i** pH value monitoring of SF-1nMgO scaffolds and SF-3nMgO scaffolds during immersion in SBF solution. Values in (**c**, **d**, **f**, **g**, **h**, **i**) represent the mean ± standard deviation (SD). Statistical difference was determined by two-tailed unpaired Student's T-test between two groups. One-way analysis of variance (ANOVA) with a Tukey's post hoc test for multiple comparisons. $n = 3$ independent replicates from three samples. Source data and exact P values are provided as a source data file. (ns: $P > 0.05$, *$P < 0.05$, **$P < 0.01$, ***$P < 0.001$).

peaks appear in those of the SF scaffolds (Fig. 2d). The FTIR spectrum and XRD patterns prove that the SF/MgO composite scaffolds were prepared.

To investigate the effect of MgO particles on the degradation behaviour of the SF scaffolds, we conducted an enzyme degradation assay of SF scaffolds and SF/MgO scaffolds at different time points (7, 14, 21, 28 d). Figure 2e showed the macroscopic appearance of the scaffolds after various degradation times. At 28 days, the SF/MgO scaffolds could better maintain structural integrity, so that the

scaffolds could also better promote interaction with cells and tissue ingrowth. Quantitative analysis of the degradation rate in Fig. 2f showed that the losing weights of SF, SF-1nMgO and SF-3nMgO were about $82.84 \pm 5.35\%$, $73.66 \pm 4.91\%$ and $61.27 \pm 6.26\%$ for 28 days, respectively. This trend implies that the degradation rate of SF scaffolds was altered according to the content of MgO particles, which proved that the degradation rate of the SF scaffolds could be regulated according to the needs of tissue regeneration. To further explore the reason for the degradation trends, we carried out zeta potential tests

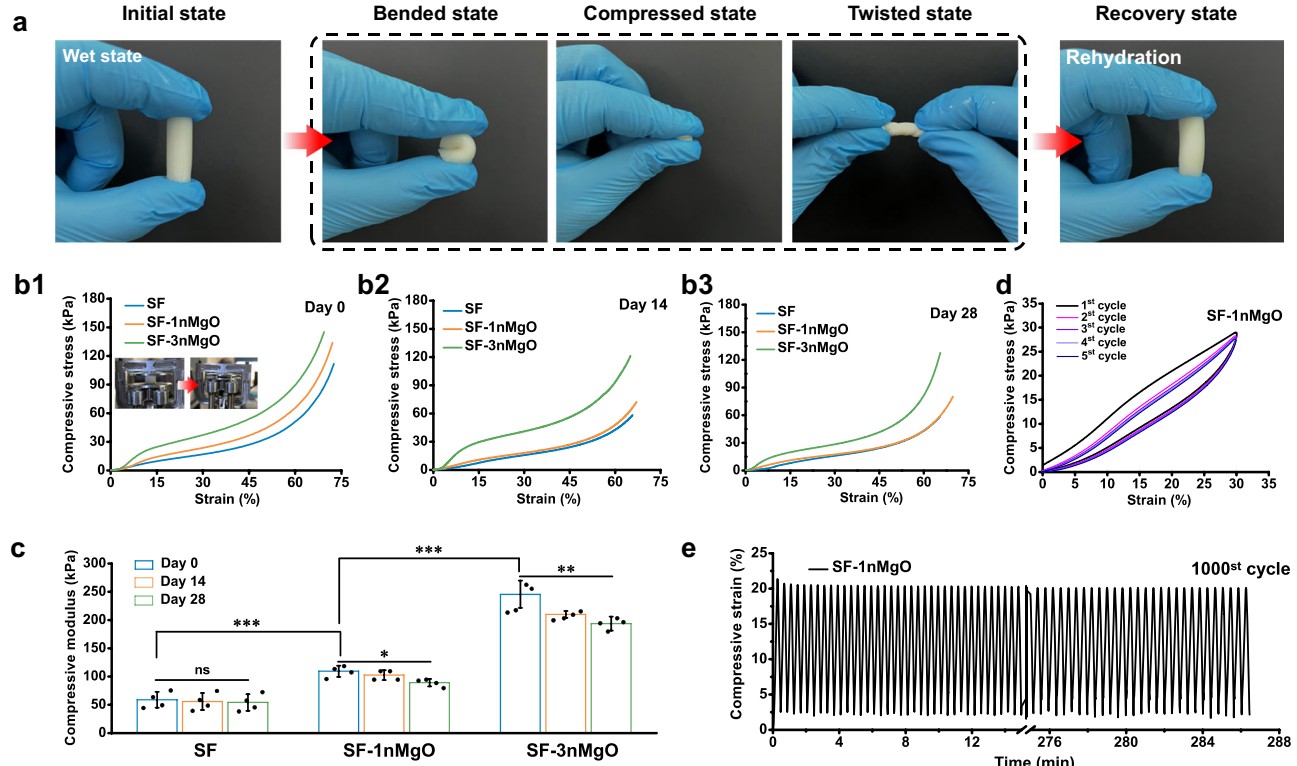

**Fig. 3 | Performances and mechanical properties of SF/MgO scaffolds.**
**a** Photographs of the SF-1nMgO scaffolds subjected to various mechanical modes, which rapidly recovered their initial shape when triggered by water. **b1-b3** Representative compressive stress-strain curves of the SF, SF-1nMgO and SF-3nMgO scaffolds. **c** Compressive modulus of the SF, SF-1nMgO and SF-3nMgO scaffolds during immersion in SBF solution after different time lengths (0, 14, 28 days). **d** Compressive stress-strain curves of the SF-1nMgO scaffolds with 30% strain for 5 cycles. **e** Compressive fatigue behaviour of the SF-1nMgO scaffolds. Values in (**c**) represent the mean ± SD. Statistical difference was determined by two-tailed unpaired Student's T-test between two groups. One-way analysis of variance (ANOVA) with a Tukey's post hoc test for multiple comparisons. $n = 4$ independent replicates from three samples. Source data and exact $P$ values are provided as a Source data file. (ns: $P > 0.05$, *$P < 0.05$, **$P < 0.01$, ***$P < 0.001$).

of the three solutions (SF, SF-1nMgO and SF-3nMgO). The results showed that the zeta potential of pure SF solutions was around -4.05 ± 0.58 mV, while the zeta potentials of the SF-1nMgO and SF-3nMgO solutions were around −7.39 ± 0.43 mV and −12.23 ± 0.46 mV, respectively (Fig. 2g). A lower zeta potential value indicates a more stable material system[43]. The accumulative $Mg^{2+}$ releasing curve and pH value of SF-1nMgO and SF-3nMgO scaffolds were measured in simulated body fluid (SBF, 37 °C) for 28 days to demonstrate the $Mg^{2+}$ releasing profile of the SF/MgO scaffolds (Fig. 2h, i). Compared to the SF-1nMgO scaffolds, SF-3nMgO scaffolds released $Mg^{2+}$ rapidly during the 4 weeks. The accumulative releasing of $Mg^{2+}$ can indirectly influence the pH value; thus, we measured the pH value of the corresponding soaking solution. The results demonstrated that the trend in pH value is consistent with $Mg^{2+}$ releasing where SF-3nMgO scaffolds are greater than SF-1nMgO scaffolds.

**Mechanical properties and shape-memory effect of bioactive silk fibroin/magnesium composite scaffolds**
The mechanical properties of the SF/MgO composite scaffolds were systematically investigated. Figure 3a showed the mechanical properties of SF/MgO composite scaffolds through the optimised preparation procedure. The obtained SF/MgO composite scaffolds could withstand complex mechanical patterns. As the applied loads were released, the SF/MgO composite scaffolds could completely recover their original shape without structural failure by water adsorption, indicating excellent mechanical stability and a perfect shape-recovery ability of the SF/MgO composite scaffolds (Supplementary Movies 1, 2). As a bone implant material, mechanical retention during degradation is essential for the formation of new bone tissue and tight integration

with the surrounding bone tissue. The mechanical properties of the scaffolds were measured during degradation. Fig. 3b1−3 illustrate the compression stress-strain curves of the scaffolds. Figure 3c displayed the results of the compressive modulus of the SF, SF-1nMgO and SF-3nMgO scaffolds. With increased MgO particle content, the compression modulus displayed an increasing trend from SF scaffolds to SF-3nMgO scaffolds, relative to 58.75 ± 14.03 kPa for SF scaffolds, 109.25 ± 9.91 kPa for SF-1nMgO scaffolds and 245.66 ± 24.21 kPa for SF-3nMgO scaffolds. The elastic modulus was between 0.5 and 0.25 MPa, which was suitable for non-load-bearing bone repair[44,45]. After 28 days of immersion in the SBF solution (pH = 7.4, 37 °C), the compressive modulus of SF/MgO composite scaffolds showed slight changes without any statistically significant differences, which could demonstrate the excellent structural stability of SF/MgO composite scaffolds during degradation. Further, taking SF-1nMgO scaffolds as an example, repeating compression-release tests demonstrated that the SF-1nMgO scaffolds could undergo multiple loading and unloading cycles with fast shape recovery, demonstrating an excellent shape-memory effect and fatigue-resistant ability (Fig. 3d, e).

The processability of bone implants during surgery is important for personalised customisation of irregularly shaped bone defects. As shown in Fig. 4a, SF/MgO composite scaffolds can be processed into large blocks that can be trimmed during surgery according to demand. The trimmed SF/MgO composite scaffolds can be implanted into defects with small calibre gaps (Supplementary Movie 3). The shape-memory property enables the SF/MgO composite scaffolds to instantly restore their original state and match the shape of the defect, achieving close integration with the bone defect. To reduce the complexity and cost of surgery, the most physiologically accessible

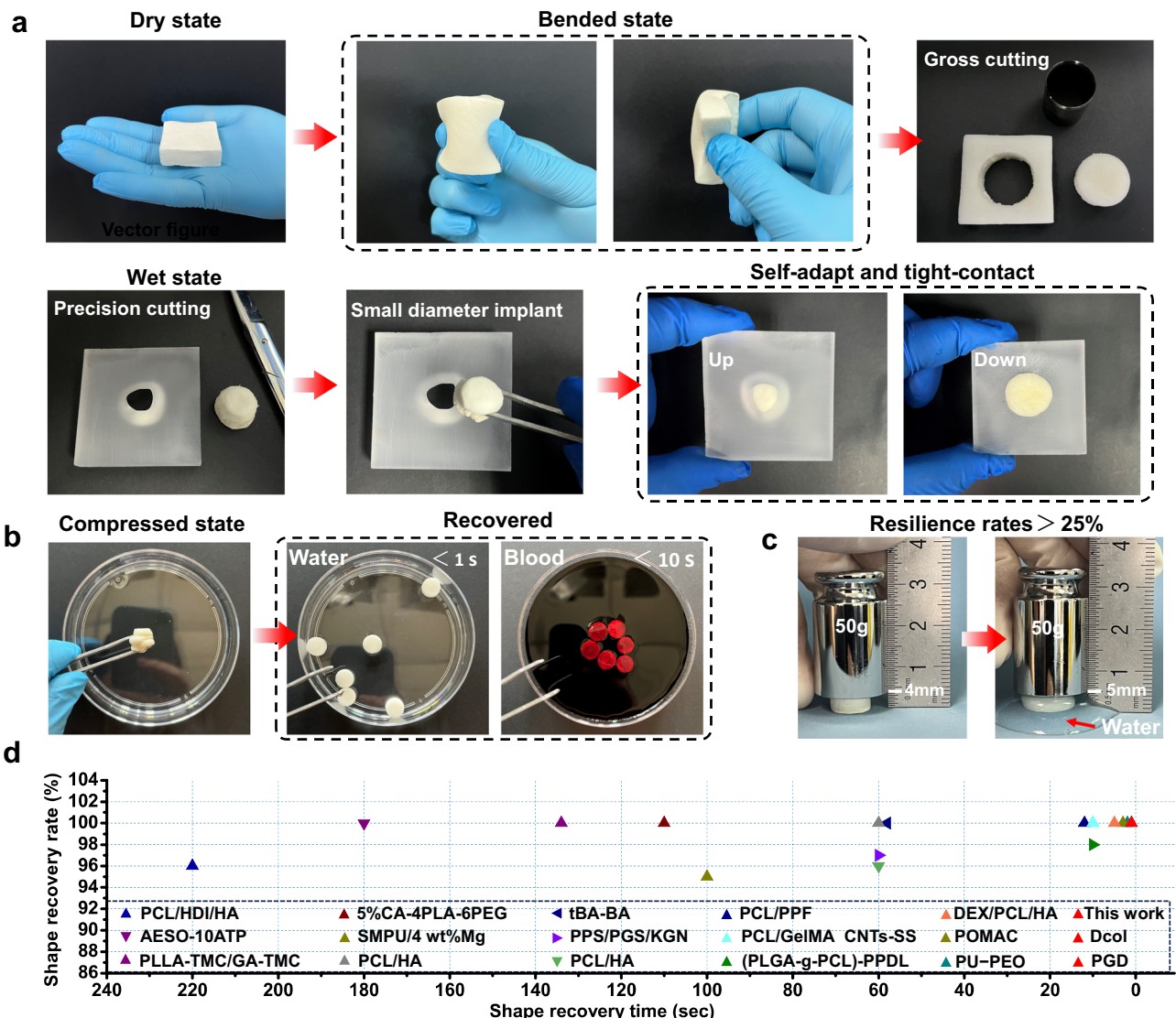

**Fig. 4 | Personalised customisation and shape-memory effect of SF/MgO scaffolds. a** The SF-1nMgO scaffolds were tailored to achieve the implantation of small pore size irregular bone defects, and the scaffolds were in close contact with the defect contour. **b** Photographs of the SF-1nMgO scaffolds, which were compressed and rapidly recovered their initial shape when triggered by water and blood. **c** The SF-1nMgO scaffolds were compressed with a 50 g weight before and after recovery when triggered by water. **d** Comparison of this work's (SF-1nMgO) composite scaffolds with other tissue engineering materials in terms of the shape-recovery rate/time. The dataset of the properties of the materials taken from the literature, with their corresponding refs. 19,20,60–76.

shape-recovery stimuli for biomedical devices are temperature and water. Figure 4b, Supplementary Fig. 5 and Supplementary Movies 4–7 provided a robust demonstration that deformed SF/MgO scaffolds have the ability to fully recover the original shape only when in contact with water or blood. The main explanation for this phenomenon falls into three categories: (i) SF has a sophisticated hierarchical structure consisting of less ordered hydrophilic (amorphous domains) and crystallisable hydrophobic blocks (β-crystals domains)[29,46]. Hydrophilic blocks provide solubility in water and are responsible for fibroin elasticity and toughness; (ii) the SF/MgO composite scaffolds are subjected to compression, and the SF amorphous domains deform first, but the structure remains intact. During this process, internal stress is generated in the scaffolds to restore the original state[47]; and (iii) domain arrangements, size and orientation of the β-crystals also influence the mechanical properties[48]. β-crystals of SF scaffolds prepared by our optimised cryogelation technology are confined to 1–2 nanometres[30], which can achieve higher strength, stiffness and toughness[48]. In addition, because of the water-responsive shape-

memory effect, the SF-1nMgO composite scaffolds could lift more than 25% at a 50 g weight (Fig. 4c). These results prove that the SF/MgO composite scaffolds have good rebound and a superior supporting ability. We also verified the thermal response characteristics of the SF/MgO scaffolds. The results showed that the deformed dry SF/MgO scaffolds cannot recover their original shape at 37 °C (Supplementary Movie 7). Therefore, it is fully confirmed that SF/MgO composite scaffolds are water-responsive shape-memory effect materials. Compared with other tissue engineering materials with shape-memory performance, SF/MgO composite scaffolds are at the forefront in terms of the shape-recovery rate and recovery time (Fig. 4d).

**In vitro biocompatibility, migration assessment and osteogenic effect of the bioactive silk fibroin/magnesium composite scaffolds**
To verify the effects of SF/MgO scaffolds on cell morphology, viability, proliferation and migration, the MC3T3-E1 pre-osteoblasts were cultured with the extract of the SF, SF-1nMgO and SF-3nMgO scaffolds

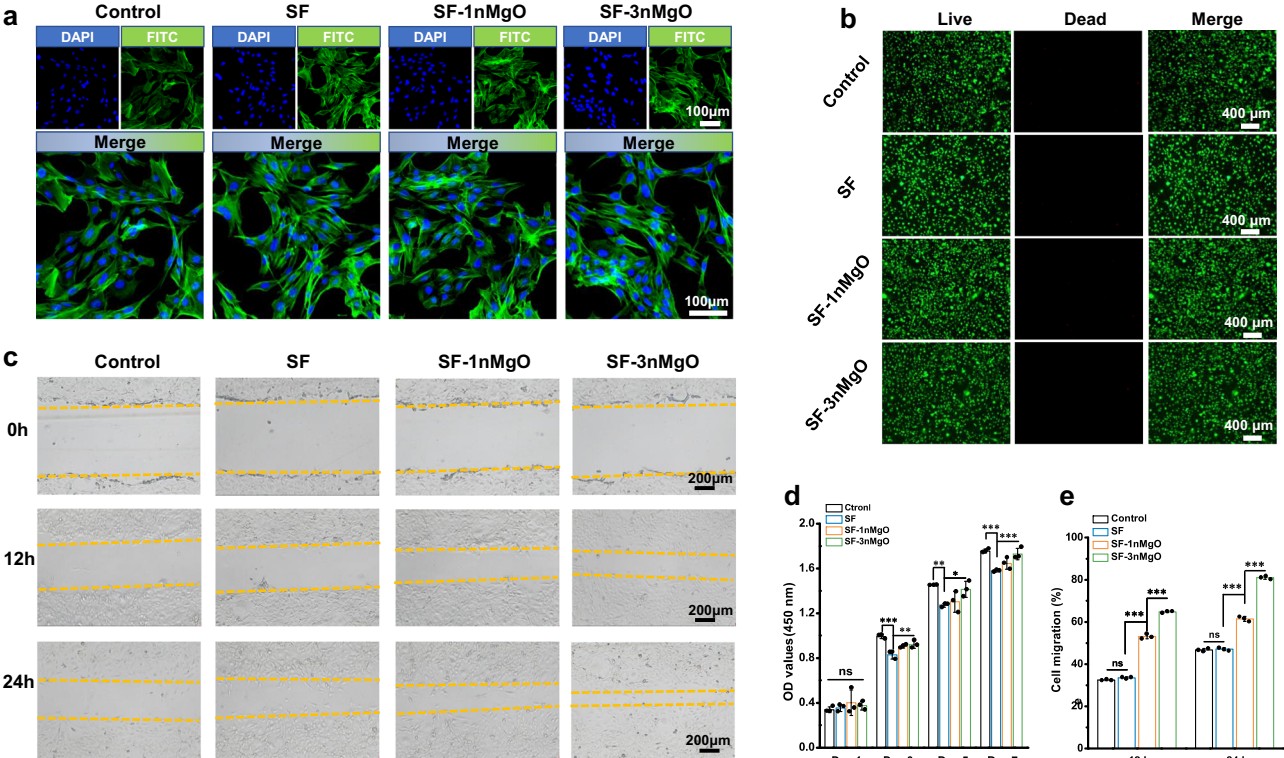

**Fig. 5 | In vitro biocompatibility and migration assessment of SF, SF-1nMgO and SF-3nMgO scaffolds. a** Attachment of MC3T3-E1 pre-osteoblasts cultured in the extract of the SF, SF-1nMgO and SF-3nMgO scaffolds observed by laser confocal microscopy. **b** Live/dead cell staining of MC3T3-E1 cells cultured in the extract of the SF, SF-1nMgO and SF-3nMgO scaffolds, where the live cells are green and dead cells are red. **c** Morphological details of the scratch of the SF, SF-1nMgO and SF-3nMgO scaffolds after treatment with extract at 0, 12 and 24 h; the yellow line represents the boundary line of cells after the scratch at different times. **d** Cell proliferation of MC3T3-E1 cells cultured in the extract of the SF, SF-1nMgO and SF-3nMgO scaffolds for 1, 3, 5 and 7 days. **e** Quantitative analysis of the migration distance in **c**. Values in (**d**, **e**) represent the mean ± SD. Statistical difference was determined by two-tailed unpaired Student's T-test between two groups. One-way analysis of variance (ANOVA) with a Tukey's post hoc test for multiple comparisons. $n = 3$ cells examined three independent experiments. Source data and exact $P$ values are provided as a Source data file. (ns: $P > 0.05$, $*P < 0.05$, $**P < 0.01$, $***P < 0.001$).

while the control group received no treatment. F-actin cytoskeleton and live/dead staining fluorescence images showed that the cells incubated with various samples displayed a well-spreading morphology and high viability (Fig. 5a, b). The CCK-8 assay indicated that the cells proliferated more over time in all groups. The optical density (OD) values of the SF-3nMgO group were significantly higher than those of the SF-1nMgO and SF groups at days 3, 5 and 7, which indicated the release of $Mg^{2+}$ in promoting cell adhesion and growth. No significant difference was observed among the SF-3nMgO and control groups in terms of OD values (Fig. 5d). The ability of the scaffolds extracts to promote cell migration was studied using a cell scratch. As shown in Fig. 5c, the results suggest that the migration ability of MC3T3-E1 cells was enhanced after treatment with SF-1nMgO and SF-3nMgO extracts compared with the SF and control groups. Quantitative analysis of the cell scratch assay showed that the scratch healing rates of the SF-1nMgO and SF-3nMgO groups reached 85.14 ± 4.50% and 95.68 ± 2.82%, respectively (Fig. 5e).

Alkaline phosphatase (ALP) staining and quantitative analysis on days 3, 7 and 14 were first performed to assess the osteogenic differentiation performance of the extraction to the BMSCs (Fig. 6a, b). At day 3, the purple colour in the stained colony of the SF/MgO group was darker than that of the control and SF scaffolds groups, and there was no significant difference between the control and SF scaffolds groups. After 7 and 14 days of incubation, the stained colony of all groups was darker, and the SF-3nMgO group still exhibited the most intense staining, followed by the SF-1nMgO, SF and control groups. The ALP activity assay showed that a twofold value was found in the SF-3nMgO group compared to the SF scaffolds group. The mineralisation

properties of BMSCs incubated with various extractions were further assessed using alizarin red S (ARS) staining (Fig. 6c, d). After 14 and 21 days of culturing, compared with the control group and the pure SF scaffolds group, the calcium nodules were significantly improved with the increase of MgO particles. The quantitative analysis of ARS showed that mineralised nodules in SF-3nMgO were around threefold higher than in the control and SF groups. Furthermore, the gene expression of the Runx2, OCN, OPN and COL I of the BMSCs after culturing for 7 and 14 days were determined by quantitative polymerase chain reaction (qPCR). Generally, all the gene expression levels were gradually upregulated over time in addition to the control group, and the related expression level in the SF-3nMgO group was significantly higher than that in the SF-1nMgO and SF groups (Fig. 6e–h). Subsequently, Western blot analysis was conducted to detect the protein levels of Runx2, OCN, OPN and COL I after 7 days of incubation. The expression levels of all genes in the SF/MgO groups significantly increased, and the SF-3nMgO group presented the highest levels of gene expression (Fig. 6i, j). These results indicated the osteopromotive effects of the SF/MgO scaffolds.

**In vivo histological and ectopic osteogenesis assessment of bioactive silk fibroin/magnesium composite scaffolds**

The foreign-body reaction (FBR) caused by the implantation of scaffolds seriously affects tissue-biomaterial integration and tissue repair[49]. Generally, in the later stage of FBR, macrophages fuse into foreign-body giant cells (FBGCs) because the materials are incapable of degrading, and the formation of fibrous capsules at the interface of materials and tissue results in the significant reduction of materials-

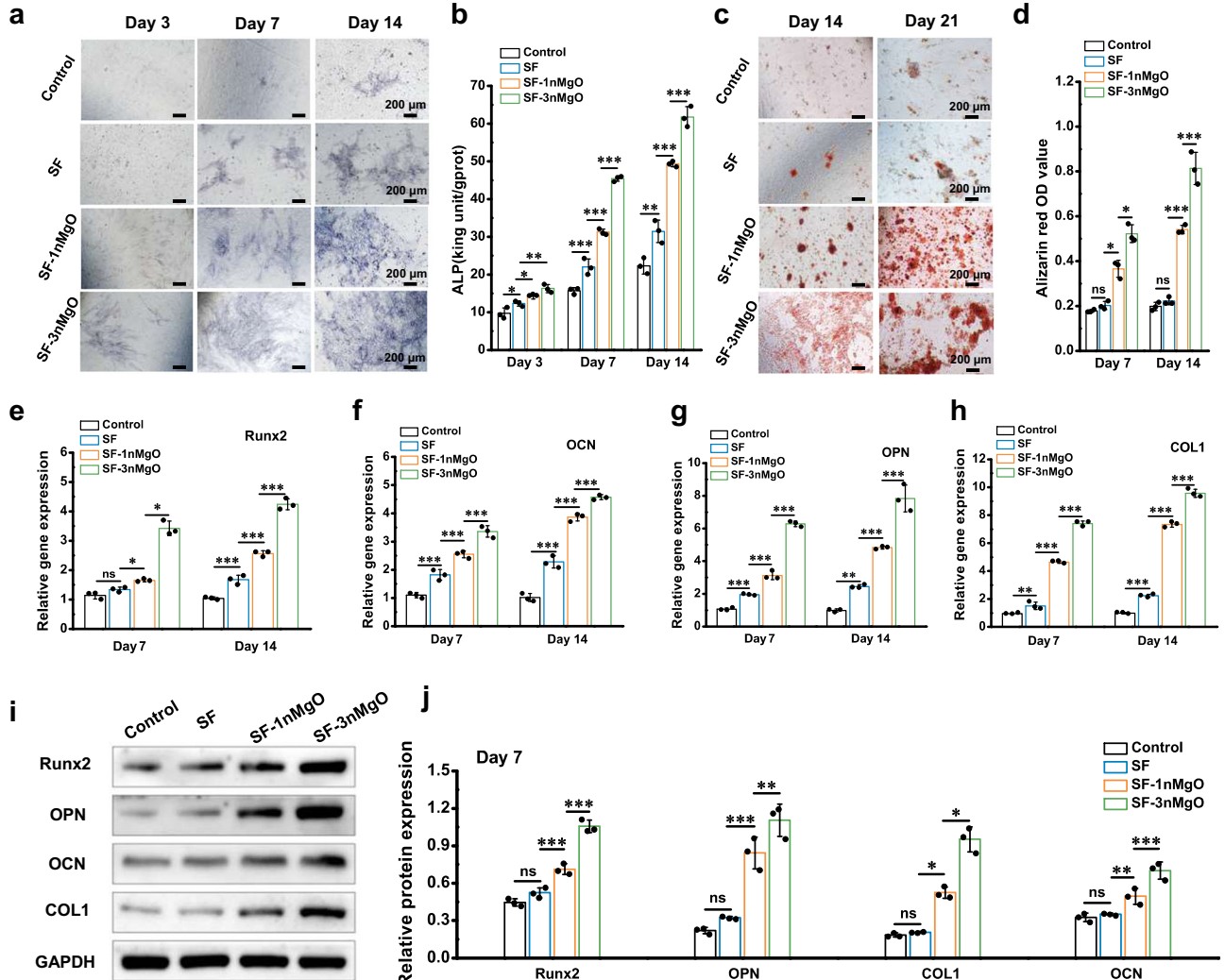

**Fig. 6 | The in vitro osteogenic effect of SF and SF/MgO scaffolds. a** ALP staining observation and (**b**) ALP activities of BMSCs cultured in the extract of the SF, SF-1nMgO and SF-3nMgO scaffolds for 3, 7 and 14 days. **c, d** ARS staining and quantitative analysis of BMSCs cultured in extracts of the SF, SF-1nMgO and SF-3nMgO scaffolds to verify the mineralisation properties. **e–h** Expression levels of bone markers including Runx2, COL1, OPN and OCN of rat BMSCs were measured and normalised to the GAPDH. **i** Western blot results of the protein expression of Runx2, OPN, COL1 and OCN after 7 days of BMSCs culture. **j** Quantitative study of the grey scale of protein expression in (**i**). Values in (**b**, **d**, **e**, **f**, **g**, **h**, **j**) represent the mean ± SD. One-way analysis of variance (ANOVA) with a Tukey's post hoc test for multiple comparisons. $n = 3$ cells examined three independent experiments. Source data and exact $P$ values are provided as a Source data file. (ns: $P > 0.05$, *$P < 0.05$, **$P < 0.01$, ***$P < 0.001$).

tissue integration. To identify the host response to scaffolds, SF, SF-1nMgO and SF-3nMgO scaffolds were implanted subcutaneously into SD rats for 4 weeks (Fig. 7a). Hematoxylin and eosin (H&E) and Masson's trichrome staining were performed to analyse host cell infiltration, the formation of FBGCs and collagenous fibrotic capsules. After 1 week of implantation, the host cells had already infiltrated across the tissue-scaffold interface into the interior of scaffolds (Supplementary Fig. 6). At 4 weeks after implantation, the SF and SF-3nMgO scaffolds exhibited less host cell infiltration; however, cell infiltration was significantly improved in the SF-1nMgO scaffolds (Fig. 7b, c). In addition, the SF and SF-1nMgO groups exhibited similar FBGC and collagenous fibrotic capsule formation, which significantly decreased compared with that of the SF-3nMgO group (Fig. 7b, d–f and Supplementary Fig. 7). These results indicated that SF-1nMgO scaffolds can effectively mitigate FBR and accelerate the integration of tissue and scaffolds.

The subcutaneous implant model in SD rats was performed to examine the ectopic osteogenesis and angiogenesis potential of the scaffolds. Figure 8a showed the OCN immunohistochemical staining results of different groups at 1, 2 and 4 weeks after surgery. The

degrees of positive expression of OCN in all experimental groups were observed at 2 and 4 weeks. At 2 weeks, SF-1nMgO and SF-3nMgO showed increased intensity compared with the SF group (Fig. 8b). At 4 weeks, the positive expression of OCN of SF-1nMgO of scaffolds increased significantly compared to the other groups (Fig. 8c). Immunofluorescence images of angiogenesis (CD31) markers showed that compared with the SF and SF-3nMgO groups, the number of CD31-positive cells representing neovascular endothelial cells was significantly improved in the SF-1nMgO group at 2 and 4 weeks (Fig. 8d–f). These results indicated that the SF-1nMgO scaffolds showed better $Mg^{2+}$-induced osteogenesis and angiogenesis ability.

## In vivo osteogenic effects of bioactive silk fibroin/magnesium composite scaffolds in a cranial defect model

Encouraged by the above results, we used the male rat critical-size calvarial defect model to characterise the therapeutic performance of the SF/MgO scaffolds in vivo (Fig. 9a). The SF-1nMgO scaffolds were selected due to their excellent histocompatibility and capacity for osteogenesis and angiogenesis. We prepared three groups: the blank

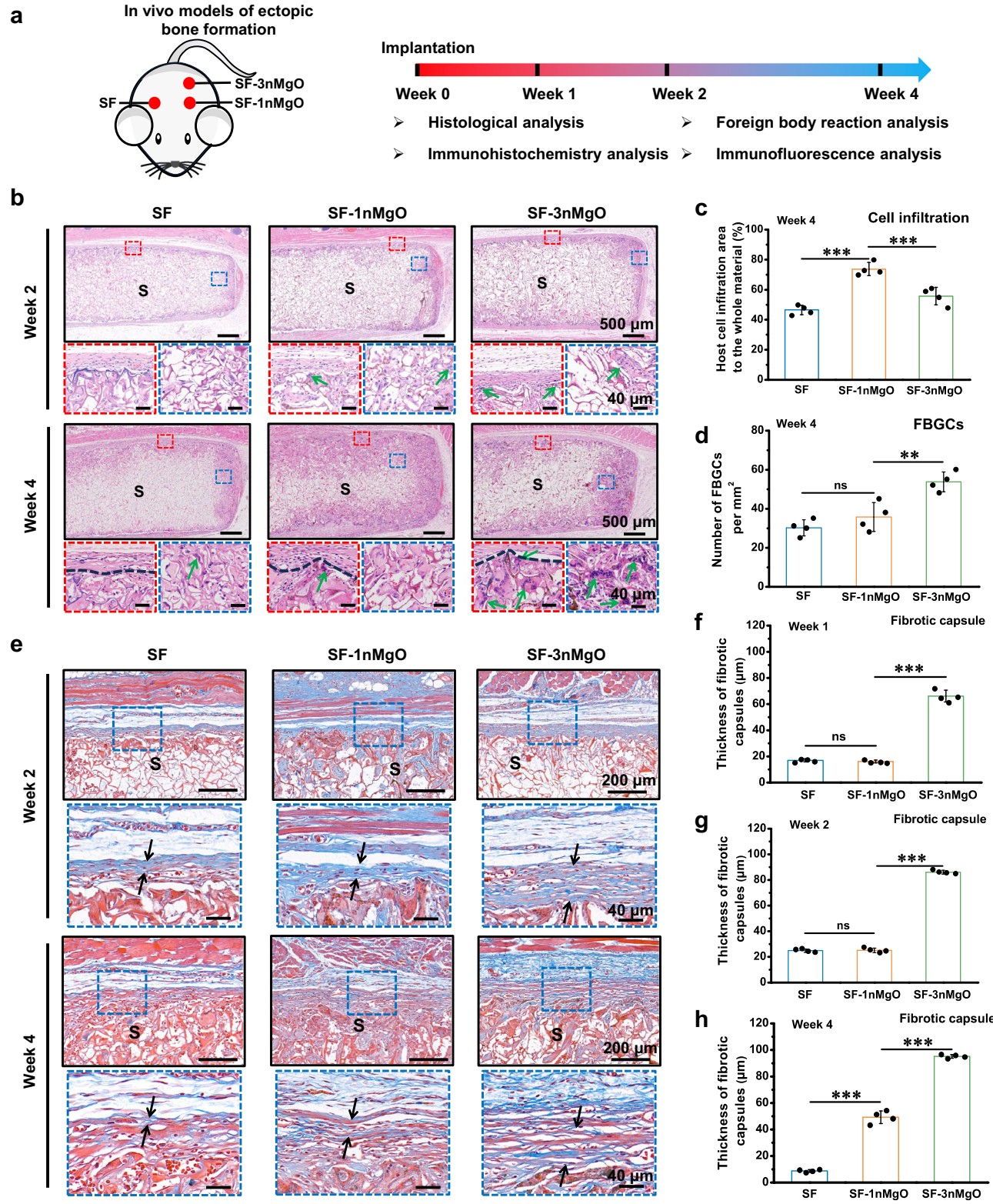

**a** In vivo models of ectopic bone formation

Implantation

Week 0 — Week 1 — Week 2 — Week 4

➢ Histological analysis        ➢ Foreign body reaction analysis
➢ Immunohistochemistry analysis  ➢ Immunofluorescence analysis

group (only defects), the control group (defects filled by the SF scaffolds) and the experimental group (defects filled by the SF-1nMgO scaffolds). First, micro-CT analysis was used to assess bone regeneration at 4 and 8 weeks after surgery. Figure 9b showed the 3D reconstruction images. Compared to the blank group and the SF scaffolds group, the SF-1nMgO group presented more newly regenerated bone at each time point, especially at 8 weeks. Moreover, for quantitative analysis, the SF-1nMgO group demonstrated enhanced bone regeneration, with the highest bone mineral density (BMD) and bone tissue volume/total tissue volume (BV/TV) values. The BMD and BV/TV of the SF-1nMgO scaffolds group were, respectively, $0.25 \pm 0.02\,\text{g/cm}^3$ and $27.2 \pm 0.64\%$ at 8 weeks, while those in the SF scaffolds group were $0.19 \pm 0.01\,\text{g/cm}^3$ and $24 \pm 1\%$, and in the blank group were only $0.08 \pm 0.01\,\text{g/cm}^3$ and $13.63 \pm 0.61\%$ (Fig. 9c, d). To further analyse the generated bone tissue within the defect area, H&E and Masson's trichrome staining were performed after 4 and 8 week implantations. The results of the H&E staining in Fig. 9e indicated that the SF-1nMgO group presented the maximum new bone tissues at each time point

**Fig. 7 | In vivo histological assessment of subcutaneous implantation in the SF, SF-1nMgO and SF-3nMgO groups. a** Experiment design, histological analysis and ectopic osteogenesis evaluation from the SF, SF-1nMgO and SF-3nMgO groups. **b** Representative images of the H&E staining of the SF, SF-1nMgO and SF-3nMgO groups after 1, 2 and 4 weeks of subcutaneous implantation; the red frame represents the boundary area between the scaffolds and the surrounding tissue; the blue frame represents the area inside the scaffolds; green arrows represent foreign body giant cells; scaffolds are labelled with S. **c** Statistical analysis of the number of host cells infiltration. **d** The number of FBGCs surrounding the implanted scaffolds 4 weeks after subcutaneous implantation. **e** Representative images of the Masson's

Trichrome staining of the SF, SF-1nMgO and SF-3nMgO groups after 1, 2 and 4 weeks of subcutaneous implantation; the blue frame represents the boundary area between the scaffolds and the surrounding tissue; the black arrow represents the fibrotic capsules; scaffolds are labelled with S. **f, g, h** Statistical analysis of the thickness of fibrotic capsules after 1, 2 and 4 weeks of subcutaneous implantation. Values in (**c, d, f, h**) represent the mean ± SD. One-way analysis of variance (ANOVA) with a Tukey's post hoc test for multiple comparisons. $n = 4$ biologically independent replicates. Source data and exact $P$-values are provided as a source data file. (ns: $P > 0.05$, *$P < 0.05$, **$P < 0.01$, ***$P < 0.001$).

---

compared to other groups, explained by known osteogenetic effects of $Mg^{2+}$. The new bone tissues grew into the pores of the scaffolds and were integrated with the scaffolds. Conversely, the control groups were mostly covered by abundant blue fibre connective tissues, and the SF group presented some fibrous tissues with a small amount of new bone tissue. We further performed Masson's trichrome staining to assess bone regeneration (Fig. 9f). More purple staining indicating mature bone could be found in the SF-1nMgO group, while the other groups showed bluer stained immature woven bone or fibrous tissues.

Bone defect healing is a coordinated process that requires a connection between blood vessels and bone cells[50]. Therefore, the success of bone regeneration also depends on vascularisation. The immunofluorescence staining of CD31 was conducted to identify the effects of scaffolds on neovascularisation (Fig. 10a, a1). The SF-1nMgO group showed more CD31 positive cells within the defect site and represented 2–4 folds of blood vessel numbers compared to the control groups and SF scaffolds at 4 and 8 weeks post-operation (Fig. 10f). In addition, immunohistochemical staining images of osteogenesis-related protein expression bone morphogenetic protein type 2 (BMP-2), Runx2, OCN and vascular endothelial growth factor (VEGF) presented similar results (Fig. 10b-e, b1–e1 and g1-4). The results showed that the highest expression level of osteogenesis-related protein appeared in the SF-1nMgO group, while the SF group presented weaker expression and the control group presented limited expression. These results demonstrate the greater potential of SF-1nMgO scaffolds to facilitate effective bone regeneration and provide a promising approach for the clinical treatment of irregular bone defects.

## Discussion

To achieve minimally invasive bone repair in clinics, in this study, a shape-memory, tailorable, self-adaptive and bioactive scaffold was developed. The optimised SF/MgO composite scaffold has an interconnected pore structure, excellent mechanical stability, water-trigger shape-memory properties and sufficient osteogenic activities. Owing to the structure stability and water-trigger shape-memory of the SF/MgO composite scaffold, the scaffolds could be compacted and implanted into the body and fully recover the original shape in contact with water or blood, and the scaffolds could provide mechanical support for tight contact with surrounding tissues and enable the inner growth of vessels and new bone in vivo. Compared with the dual-crosslinked SF scaffolds, the mechanical performance of the multi-crosslinking SF/MgO composite scaffolds increased significantly after incorporating the nano-MgO particles; this design successfully overcame the malleability and brittleness of the calcium phosphate bone (CPC) scaffolds, which is a popular injectable biomaterial in orthopaedic surgeries. Moreover, for bone implants, mechanical retention during materials degradation directly affects the integration ability of the material-tissue interface[23]. With the increase of MgO partials, the degradative rate of the SF/MgO scaffolds gradually slowed, and excellent mechanical retention was well maintained during the degradative process. Therefore, in this SF/MgO scaffold, the degradative rate and mechanical stability could be controlled by adjusting the amount of MgO particles.

Previous studies have shown that $Mg^{2+}$ plays an important role in the regulation of the cellular functions of pre-osteoblasts and BMSCs, such as protein synthesis, activation and bone formation[51,52]. To avoid the generation of hydrogen in Mg metal biomaterials during rapid corrosion in an aqueous environment, MgO, as the source of magnesium element, was widely used in conjunction with other materials[38,53]. Our group previously proved that an appropriate dosage range of $Mg^{2+}$ level could promote the proliferation, viability, adhesion, migration and osteogenic differentiation of osteoblasts[40,54]. In this study, the results of in vitro experiments demonstrated that Mg-contained SF/MgO composite scaffolds (SF-1nMgO and SF-3nMgO) not only significantly enhance the attachment, proliferation, viability and migration of MC3T3-E1 pre-osteoblasts, but can also promote the osteogenic differentiation of BMSCs and accelerate angiogenesis-related factor expression.

Studies have shown that the rapid degradation of materials may lead to the loss of mechanical support and eventually incomplete tissue formation within materials[55,56]. By contrast, slow degradation of the material could cause severe FBR, namely the formation of a thick collagen fibrotic capsule, which greatly reduces the interface integration of the materials and the host tissues[57–59]. In this study, in vitro physicochemical properties studies showed that an increase in MgO particle content led to a more stable structure of SF/MgO composite scaffolds, and the degradation rate decreased significantly. The results of subcutaneous implantation demonstrated that the collagenous fibrous capsule thickness of the SF-1nMgO scaffolds was lower than that of the SF-3nMgO scaffolds. Nevertheless, it was found that the structure of the fibrous capsule changed from dense to loose, showing a tendency of ablation, which would be highly beneficial for the infiltration of host cells and tissue-biomaterial integration. In addition, the subcutaneous ectopic osteogenesis results confirmed that SF-1nMgO scaffolds with a suitable degradation rate formed a thin collagenous fibrous capsule, which enriched the interfacial tissue microenvironment and showed better $Mg^{2+}$-induced osteogenesis and angiogenesis ability. The qualitative and quantitative results of subcutaneous implantation revealed that SF-1nMgO scaffolds have a high potential to stimulate bone repair for the repair of critical-size bone defects.

After the cell-free SF-1nMgO scaffolds were implanted in the rat cranial defects, micro-CT, histological staining, immunohistochemistry and immunofluorescence analysis suggested that the defective bone sites in SF-1nMgO bioactive scaffolds promoted bone regeneration more than those in the blank and SF scaffolds groups, which might be caused by the bioactive effects of released $Mg^{2+}$. The results of osteogenesis in vivo were consistent with those of in vitro experiments and subcutaneous implantation. Additionally, the SF-1nMgO scaffolds were confirmed to have enhanced biomechanical properties and tight contact with the defective site for robust bone regeneration. After 4 and 8 weeks of implantation, higher osteo-related expression of BMP-2, Runx2 and OCN in SF-1nMgO scaffolds was confirmed. The higher positive expression level of VEGF in SF-1nMgO scaffolds demonstrated greater potential for vascularisation. In vivo cranial bone repair generally took over 8 weeks, indicating that the degradation duration of scaffolds should meet the requirement of the healing time. At 8 weeks after SF-1nMgO scaffolds implantation, the histological staining results

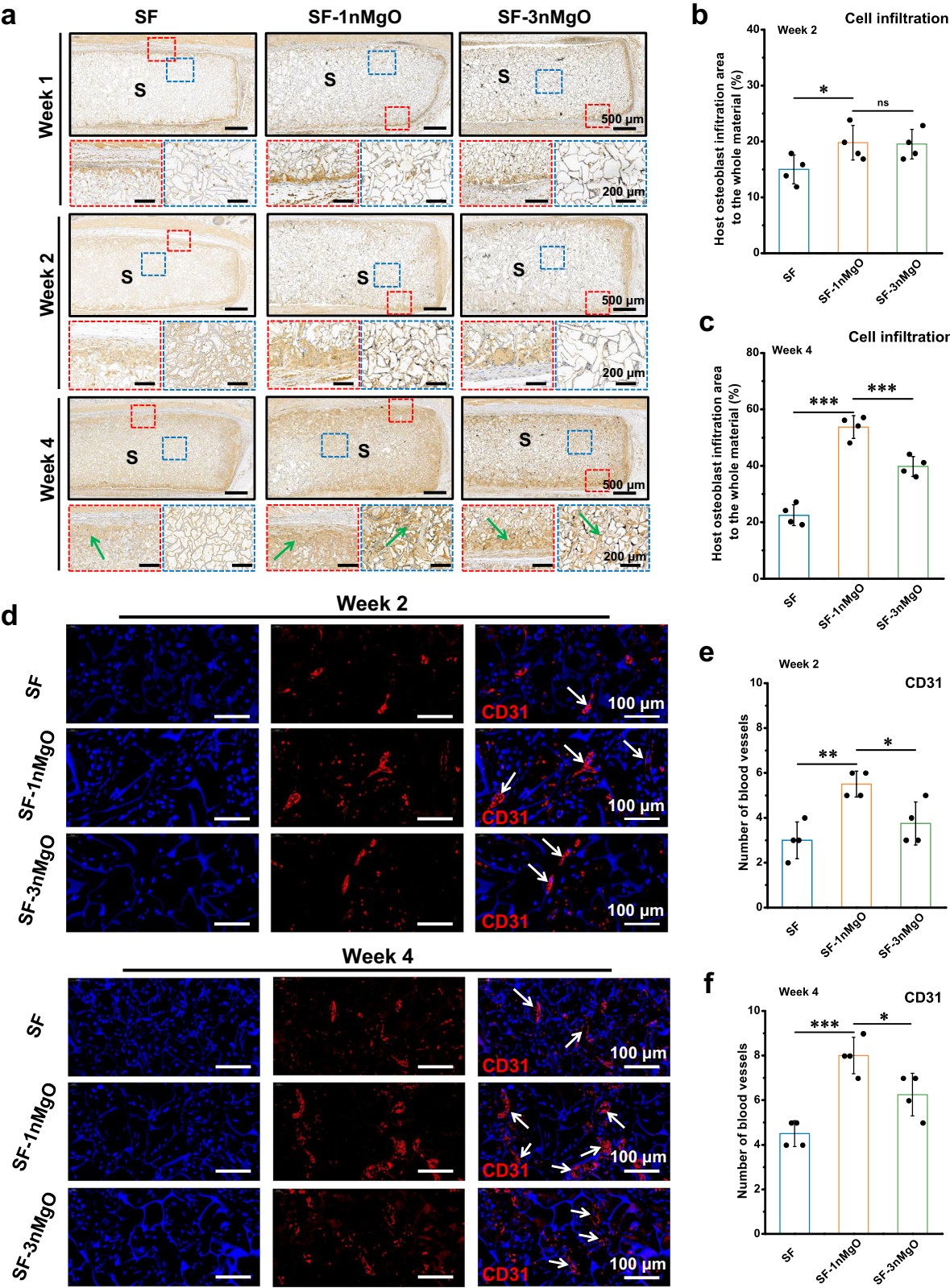

**Fig. 8 | Evaluation of ectopic osteogenesis in vivo in the SF, SF-1nMgO and SF-3nMgO groups. a** OCN staining images of different groups at 1, 2 and 4 weeks after the surgery; the red frame represents the boundary area between the scaffolds and the surrounding tissue; the blue frame represents the area inside the scaffolds; scaffolds are labelled with S; green arrows: new bone tissues. **b**, **c** Statistical analysis of the number of host cells infiltration after 2 and 4 weeks of subcutaneous implantation. **d** Immunofluorescence staining of platelet endothelial cell adhesion molecule-1 (CD31) at 2 and 4 weeks after surgery; the white arrows represent newly formed blood vessels. **e**, **f** Statistical analysis of blood vessels within the scaffolds at 2 and 4 weeks post-operation; white arrows: new blood vessels. Values in (**b**, **c**, **e**, **f**) represent the mean ± SD. One-way analysis of variance (ANOVA) with a Tukey's post hoc test for multiple comparisons. $n = 4$ biologically independent replicates. Source data and exact $P$ values are provided as a Source data file. (ns: $P > 0.05$, *$P < 0.05$, **$P < 0.01$, ***$P < 0.001$).

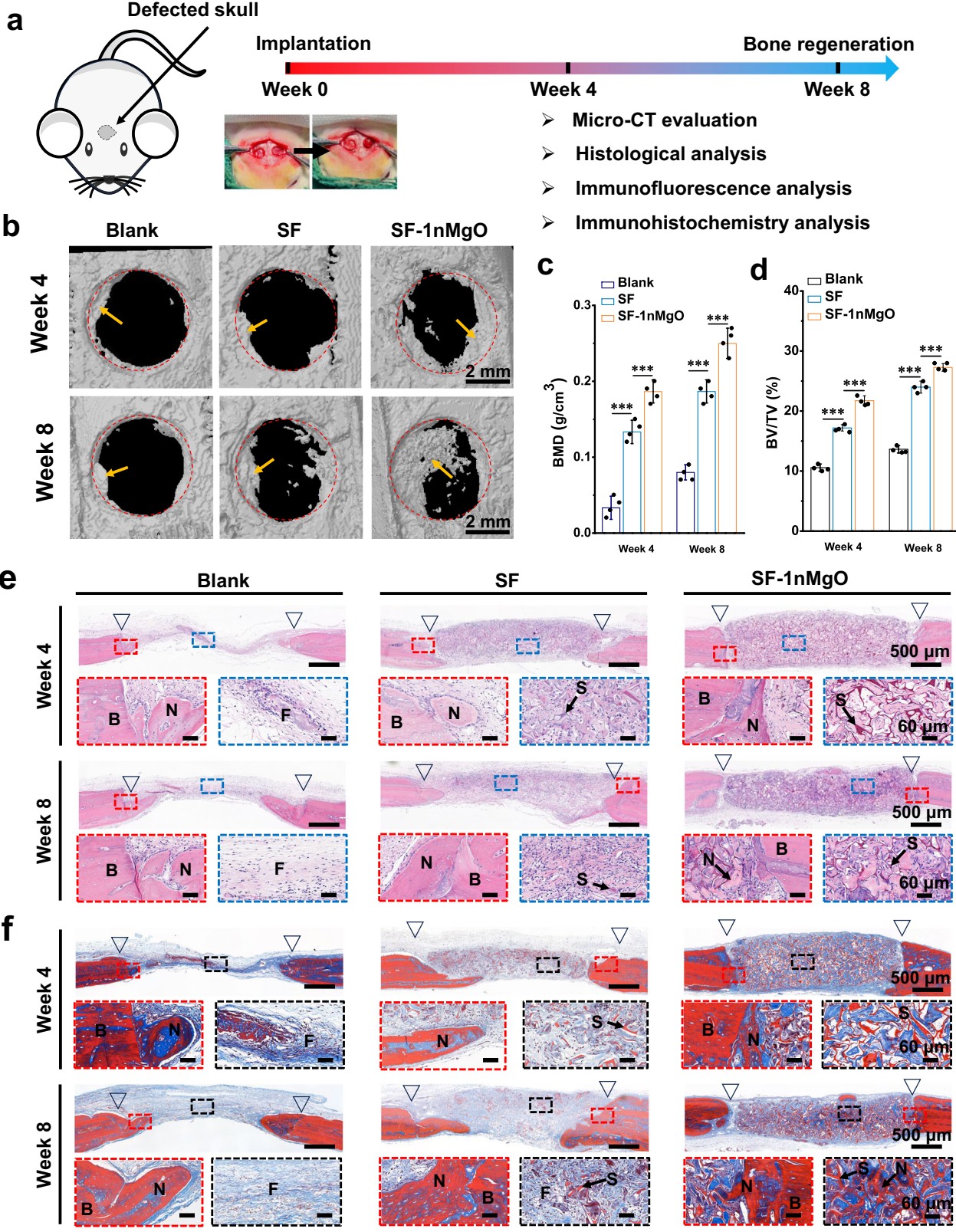

**Fig. 9 | In situ skull regeneration enhanced by SF/MgO scaffolds in a rat cranial defect model. a** Experiment design includes micro-CT reconstruction, histological analysis and osteogenesis evaluation from the SF and SF-1nMgO groups. **b** Micro-CT reconstructed images of defective bone at 4 and 8 weeks within the blank, SF and SF-1nMgO groups; red circle: defective area; yellow arrows: new bone tissues. **c** BMD and (**d**) bone volume change (BV/BT) varied in the blank, SF and SF-1nMgO groups at 4 and 8 weeks. **e** H&E and (**f**) Masson's trichrome staining and magnified views of bone formation within the cranial defects within the blank, SF and SF-1nMgO groups after implantation for 4 and 8 weeks (N new bone tissue. S scaffolds. F fibrous tissue. B old bone boundary). The triangular symbol indicates the initial boundary of the defect. Blank: defect alone with no treatment. Values in (**c, d**) represent the mean ± SD. One-way analysis of variance (ANOVA) with a Tukey's post hoc test for multiple comparisons. *n* = 4 biologically independent replicates. Source data and exact *P* values are provided as a Source data file. (****P* < 0.001).

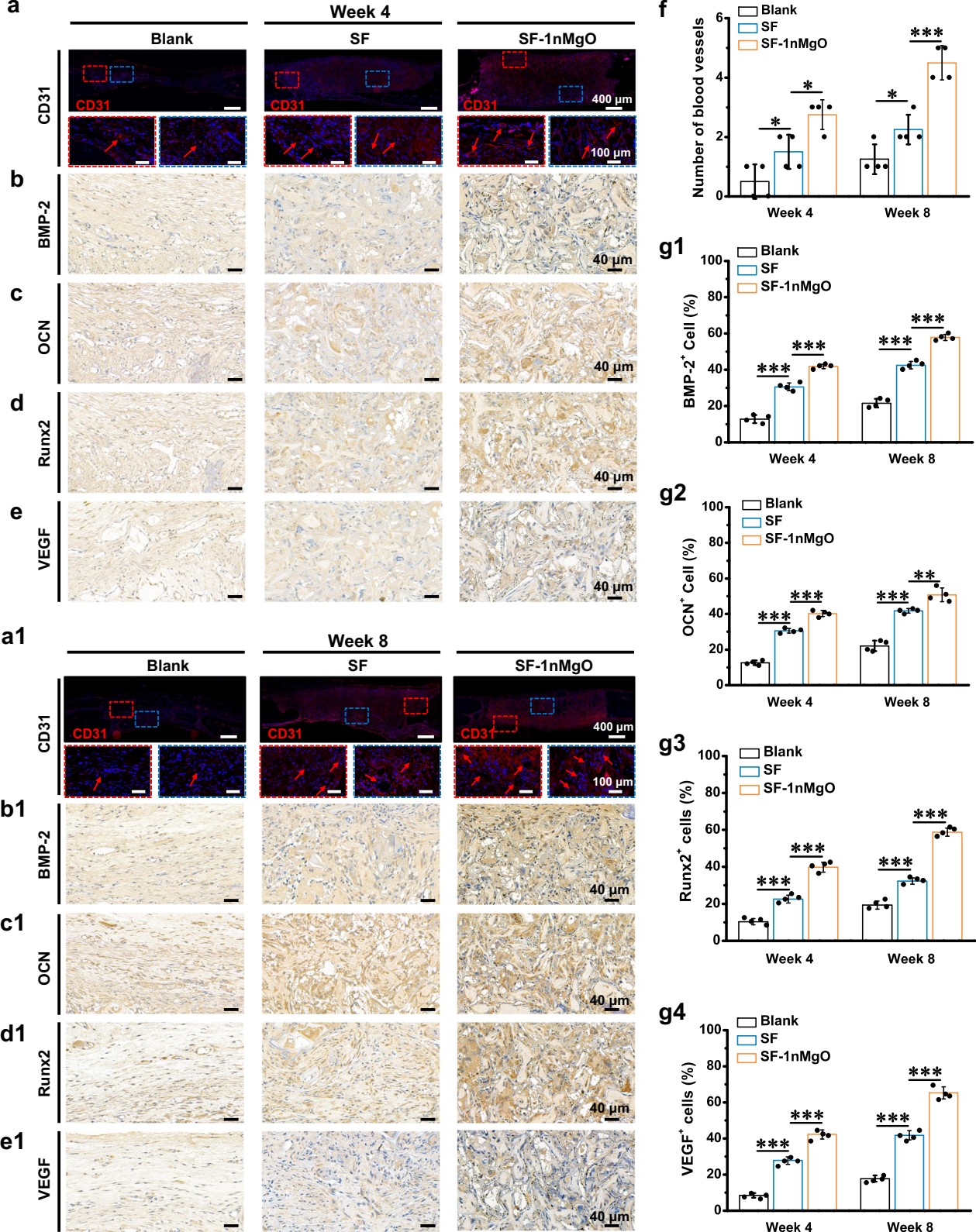

**Fig. 10 | Representative staining images of regenerated tissue to the blank, SF and SF-1nMgO groups at 4 and 8 weeks. a, a1** CD31 immunofluorescence staining of regenerated vessels at weeks 4 and 8; new blood vessels (red arrow); the red frame represents the boundary area between the scaffolds and the surrounding tissue; the blue frame represents the area inside the scaffolds. **b–e** and **b1–e1** Representative immunohistochemistry images of BMP-2, OCN, Runx2 and VEGF at 4 and 8 weeks after implantation. **f** Quantitative analysis of blood vessels within the scaffolds at 4 and 8 weeks post-operation. **g1–g4** Quantitative analysis of positive cells at 4 and 8 weeks. Blank: defect alone with no treatment. Values in (**f, g1, g2, g3, g4**) represent the mean ± SD. One-way analysis of variance (ANOVA) with a Tukey's post hoc test for multiple comparisons. $n = 4$ biologically independent replicates. Source data and exact $P$ values are provided as a Source data file. (*$P < 0.05$, **$P < 0.01$, ***$P < 0.001$).

showed partial degradation of the SF-1nMgO scaffolds, accompanied by in situ regeneration of new bone tissue inside the scaffolds. Therefore, a prolonged implantation time of the SF-1nMgO scaffolds meant the remaining MgO inside the degrading scaffolds would gradually hydrolyse and release $Mg^{2+}$ to promote bone repair. By contrast, SF scaffolds were largely degraded after 8 weeks of implantation, but very limited new bone formation was seen. Therefore, while the addition of MgO particles to SF scaffolds enhanced the biological activity, it also brought more possibilities for regulating the degradation rate of scaffolds for bone repair.

Taken together, the water-/blood-responsive shape-memory effect enables the multi-crosslinked cell-free SF-1nMgO composite scaffolds to instantly restore their original state and match the shape of the defect, achieving tight contact with the bone defect. The degradation of SF/MgO composite scaffolds is accompanied by the gradual precipitation of $Mg^{2+}$, which enriches the interfacial tissue microenvironment to promote pre-osteoblasts and BMSCs migration towards the bone defect. Furthermore, $Mg^{2+}$ regulates the functions of migrating cells, including adhesion, proliferation and vascularisation, and it upregulates the expression of osteogenic-related genes (Runx2, OCN, OPN and COL I), which finally promotes in situ irregular bone regeneration.

There are still limitations to our study. The compression modulus of the SF/MgO composite scaffolds is between 0.1 and 0.3 MPa, which is suitable for non-load-bearing bone repair; an irregular-shaped bone defect model should be established that could fully demonstrate and reflect the performance of the SF/MgO composite scaffolds. The time spent studying animal models needs to be extended to evaluate the long-term bone repair effects of SF/MgO composite scaffolds. In future studies, a large experimental animal model should be established to further verify the concept of minimally invasive treatment of irregular bone defects and to explore the related molecular mechanism of osteogenesis.

## Methods

We have complied with all relevant ethical regulations declared in the manuscript, and disclosed the name(s) of the board and institution in the methods part. All procedures were approved by the animal care and use committee of Beijing Jishuitan Hospital regarding the Guiding Principles for the care and use of experimental animals (Number: 2022-12-03).

### Materials

*Bombyx mori* cocoons were procured from Favorsun Medical Technology (Shanghai) Co., Ltd., China. Lithium bromide (LiBr, 99%), EGDE, Epoxy value ≥ 0.7, N,N,N',N'-Tetramethylethylenediamine (TEMED, 99%), dialysis tubes (8000–14,000 MWCO) and dialysis tubes (3500 MWCO) were obtained from Sigma-Aldrich, United States. Polyethylene glycol (PEG, Mn ~ 100,000 g mol[-1]) and sodium bicarbonate (NaHCO$_3$, 99.8%) were purchased from Macklin Chemical Reagent Co., Ltd., China. MgO particles (50 nm, 99%) were obtained from Aladdin, Shanghai. Simulated body fluid (SBF, pH = 7.4) and phosphate buffered saline (PBS, pH = 7.4) were purchased from Beijing Solarbio Science & Technology Co., Ltd.

### Preparation of silk fibroin solution

The SF solution was extracted from *Bombyx mori* cocoons. Briefly, silk cocoons (10 g) were degummed in a boiling NaHCO$_3$ solution (2 L, 5 g/L) for 1 h, and sericin was removed with water. The degummed silk fibres were oven-dried at 45 °C for 24 h. Then, the dry degummed silk fibres (8 g) were dissolved in LiBr solution (100 mL, 9.3 M) for 3 h at 40 °C. The SF solution was subsequently dialysed (MWCO 8000–14,000) against deionized water for 3 days; and the deionized water was changed 3 times a day to remove ions and impurities. The purified SF solution (~20 mg/mL) was acquired. Finally, the highly concentrated SF solution (50–60 mg/mL) was obtained by concentration in 15% PEG solutions using dialysis membrane (3500 MWCO).

### Fabrication of silk fibroin/magnesium composite scaffolds

The silk fibroin/magnesium composite scaffolds were synthesised through cryogelation. The process could be found in Supplementary Fig. 1. Briefly, SF solutions (60 mg/ml), EGDE (2 mmol/g) and TEMED (0.25 v/v%) were added in a centrifuge tube. Next, MgO nanoparticles were added to deionised water to prepare a 90 mg/mL suspension. MgO suspension was added to the above SF mixture solution according to the ratio of the solute mass of the SF solution to the mass of the MgO nanoparticles at 10:0 (SF), 10:1 (SF-1nMgO) and 10:3 (SF-3nMgO), respectively. The above solution was uniformly mixed, and the cryogel reaction was performed at −10 °C for 24 h. After thawing in a water bath (room temperature) for 12 h, SF/MgO composite scaffolds were obtained. The residual crosslinkers and catalysts in SF/MgO scaffolds were removed by rinsing with ultrapure water. The SF/MgO scaffolds were frozen at −20 °C for 12 h, followed by freeze-drying at -50 °C for 24 h. The dry scaffolds were then sterilised using cobalt-60 (25 kGy) radiation for 2 h.

### Characterisation of silk fibroin/magnesium composite scaffolds

The pore size and micromorphology of SF/MgO scaffolds were observed by scanning electron microscopy with an energy-dispersive spectrometer (SEM-EDS, S-4800, Hitachi, Japan) at an accelerating voltage of 10 kV. The cryogeled SF/MgO composite scaffolds were first frozen at −20 °C for 12 h, and vacuum-dried at −50 °C for 24 h. The dry SF/MgO scaffolds were sputter-coated with gold for the SEM test.

The wide-angle X-ray diffraction (XRD) patterns of the SF/MgO scaffolds were conducted using an XRD (D8 Advance, Bruker, Germany) and a Cu Ka radiation source (1.54 Å). The SF/MgO scaffolds were scanned from 2θ = 10° to 60°. The surface chemical and conformation analysis of the SF/MgO scaffolds were measured by attenuated total reflection Fourier-transform infra-red spectroscopy (ATR-FTIR, Thermo Scientific Nicolet iS20) over a wavenumber range of 400 to 4000 cm[-1]. 32 scans with a resolution of 4 cm[-1] were accumulated for each spectrum at 25 °C. The conformation contents of the SF/MgO scaffolds in the amide I region (1595–1705 cm[-1]) was analyzed by Fourier self-deconvolution (FSD) using PeakFit software (version 4.12)[42]. The 1–2 mm thick slices of freeze-dried scaffolds were compacted for the XRD and FTIR tests.

### Degradation behaviour of silk fibroin/magnesium composite scaffolds

The dry scaffolds were prepared using 1 ml of homogeneous reaction solution (SF, SF-1nMgO, SF-3nMgO) by cryogel reaction and used as samples for degradation experiments. The SF, SF-1nMgO and SF-3nMgO scaffolds were incubated at 37 °C in a PBS solution that contained 2 U/mL protease (Type XIV from Streptomyces griseus (3.5 U/mg)). The enzyme solutions were changed every 2 days to maintain the enzyme activity. The scaffolds were removed from the enzyme solution and rinsed with deionised water at indicated time (3, 7, 14 and 28 days), freeze-dried and weighed. Weight and morphology were recorded. The weight loss (%) of the scaffolds was calculated as follows:

$$\text{Weight loss}(\%) = (W_0 - W_t)/W_0 \times 100\%, \quad (1)$$

where $W_t$ and $W_0$ are the weights of the remaining and initial scaffolds at different time points, respectively.

### In vitro $Mg^{2+}$ release behaviour of silk fibroin/magnesium composite scaffolds

The immersion test was performed to measure the accumulative release profile of magnesium ion ($Mg^{2+}$) from the SF/MgO composite

scaffolds. The samples were prepared using 1 ml of homogeneous reaction solution (SF-1nMgO, SF-3nMgO) using a cryogel reaction. The two samples (SF-1nMgO and SF-3nMgO) were incubated individually in 2 mL of SBF (pH = 7.4) at 37 °C. The $Mg^{2+}$ concentrations in the extract liquids of the samples were measured by inductively coupled plasma-optical emission spectrometry (ICP-OES; Thermo Fisher Scientific, USA) at 3, 7, 14 and 28 days, respectively. The pH change of the extract liquids was measured using a pH metre (Mettler, Switzerland) and recorded.

## Zeta potential measurement

The zeta potential of the SF solution, SF-1nMgO solution (SF: MgO, 10:1, w/w) and SF-3nMgO solution (SF: MgO, 10:3, w/w) solution was measured at 25 °C (Malvern Zetasizer Nano ZS90) according to ISO-13099-2. The sample concentration was diluted to one-half of the initial concentration for testing. Three measurements were obtained for each group.

## Mechanical properties of silk fibroin/magnesium composite scaffolds

The mechanical retention during degradation and cyclic compression properties of the SF and SF/MgO scaffolds were measured using a dynamic mechanical analyser (DMA) Q800 (TA Instruments, Waters Ltd. USA). The compression testing samples were cut to 10 mm × 10 mm (diameter × height). The mechanical retention of samples was incubated at 37 °C in SBF solution and removed at indicated time points (0, 14 and 28 days). On day 0, all samples were adequately hydrated in PBS for 12 h before testing. The compression test was carried out at a strain rate of 30%/min at room temperature. Young's modulus of samples was calculated from the initial linear strain range of the curve obtained from the stress-strain. Five cycles compression testing of the SF-1nMgO scaffolds was measured at a strain rate of 30%/min with a maximum strain of 30%. Fatigue-resistant ability testing of the SF-1nMgO scaffolds was conducted between −2% and −22% strain at a strain rate of −100%/min for 1000 cycles.

## Shape-memory effect and personalised customisation of silk fibroin/magnesium composite scaffolds

The SF/MgO scaffolds were fully hydrated in PBS solution and subsequently subjected to bending, compression and torsion deformation, followed by exposure to room-temperature water to evaluate their ability to restore their original shape. Multiple scaffolds were completely compressed together and subsequently exposed to room-temperature water and blood to record the rate and extent of recovery of the scaffolds to their original shape in different media environments. In addition, the compressed wet SF-1nMgO scaffolds were placed at a weight of 50 g. The scaffolds were then incubated in room-temperature water and the images were recorded. Furthermore, the potential of SF-1nMgO scaffolds to be trimmed during clinical surgery for personalised implantation was explored. The bulk SF-1nMgO scaffolds were prepared and then roughly and precisely trimmed according to the shape of the bone defect. Finally, the SF-1nMgO scaffolds were wetted and compressed to achieve the implantation of small pore-size irregular bone defects.

## In vitro cell biocompatibility and migration of silk fibroin/magnesium composite scaffolds

Mouse embryo osteoblast precursor (MC3T3-E1) cells (PUMC000012, Cell Resource Center, IBMS, CAMS/PUMC) were utilised for evaluating cell viability, proliferation and migration. The extract liquid of the SF, SF-1nMgO and SF-3nMgO scaffolds was prepared according to ISO 10993-12. First, the scaffolds were sterilised using cobalt-60 (25 kGy) radiation for 2 h. Then, the sterilised scaffolds with a fixed volume to medium volume ratio (1.25 cm² ml⁻¹) were immersed in Dulbecco's modified Eagle's medium (DMEM, high glucose, Invitrogen, USA)

supplemented with 10% (v/v) foetal bovine serum (FBS, Gibco, USA) and 1% (v/v) penicillin/streptomycin (Gibco, USA) (30 ml in total) for 24 h at 37 °C in 5% $CO_2$ chamber. For live/dead staining, MC3T3-E1 cells were incubated with the extraction in a 24-well culture plate with a density of $2 \times 10^4$ cells/well in an incubator under 37 °C and 5% $CO_2$ for 3 days. The viability of the cells was assessed using a live/dead viability/cytotoxicity kit (Thermo Fisher, USA). The fluorescence photograph was obtained using confocal laser scanning microscopy (CLSM, Leica SP8, Germany). To determine cell proliferation, a Cell Counting Kit-8 (CCK-8, Dojindo, Japan) assay was conducted after being incubated for 1, 3, 5 and 7 days according to the manufacturer's instructions. The optical density at 450 nm (OD450) was measured using a spectro-photometer (ThermoFisher, USA). The spreading morphology of the cells cultured with the extraction was determined using CLSM. After being fixed with 4% paraformaldehyde, the cells were treated with 0.2% Triton X-100 (Sigma, USA) in PBS. After rinsing, the cells were stained with phalloidin-FITC (1:200 dilution, Solarbio, China) for 30 min and the mounting medium (with DAPI) (Solarbio, China) was used to cover the slides for CLSM observation. A scratch assay was performed to investigate the effect of scaffolds on the migration of MC3T3-E1 cells. When the fusion rate of the MC3T3-E1 cells reached 90%, a cell scratch was prepared using the tip of a 200 µl pipette. After rinsing with PBS, the scaffold extracts were added to the well and cultured at 37 °C. The photographs were obtained at 12 and 24 h using a light microscope. Cell migration images were analysed using ImageJ software. The cell migration ratio was calculated using the following equation:

$$\text{Cell migration ratio}(\%) = (L_0 - L_t)/L_0 \times 100\%, \qquad (2)$$

where $L_0$ and $L_t$ are the scratched widths before and after the addition of the extract, respectively.

## In vitro osteogenic differentiation study of silk fibroin/magnesium composite scaffolds

Bone mesenchymal stem cells (CP-R131, Procell Life Science & Technology Co,. Ltd) were utilised to evaluate the osteogenic differentiation. The bone mesenchymal stem cells (BMSCs) were cultured in a fresh osteogenic induction medium (ascorbic acid 50 µg/mL, dexamethasone 100 nM, β-glycerol phosphate 10 mM). The culture plate was incubated in a 5% $CO_2$ chamber at 37 °C. The extract liquid of the SF, SF-1nMgO and SF-3nMgO scaffolds was prepared as mentioned above. Alkaline phosphatase (ALP), ALP activity, Alizarin Red S (ARS) staining, quantitative polymerase chain reaction (qPCR), and western blot analysis were conducted to assess the osteogenic differentiation of the BMSCs. ALP activity was detected with p-nitrophenyl phosphate (pNPP) (Beyotime, China) at 3, 7 and 14 days after the addition of scaffold extracts to the BMSCs. ALP activity was quantified by absorbance measurements at 405 nm. The mineralisation of the BMSCs was identified by ARS solution (Solarbio, China) according to the manufacturer's protocol after 14 and 21 days. Images were photographed using an optical microscope (Nikon, Japan). The amount of calcium deposition was further investigated quantificationally.

The gene expression of osteogenic differentiation markers runt-related transcription factor 2 (Runx2), osteocalcin (OCN), osteopontin (OPN) and collagen I (COL I) were evaluated through qPCR after the cells were cultured for 7 and 14 days. The total RNA of the cells was extracted using TRIzol reagent (15596026CN, Invitrogen, USA) and reverse-transcribed to complementary DNA (cDNA) using a PrimeScript RT kit (Takara, Tokyo, Japan). SYBR Green Master Mix (Roche Applied Science, Germany) was used to perform real-time PCR. The primer pairs used are shown in Supplementary Table 1. The relative mRNA expression levels of genes were normalised to the GAPDH using CT values. For western blot analysis, after 14 days of culture, the total proteins from BMSCs were lysed with Radio Immuno Precipitation Assay (RIPA) and the concentration of protein was measured

using the BCA protein assay kit (Thermo Fisher Scientific, USA). After that, the proteins were boiled and separated by SDS-PAGE followed by transformation to PVDF membranes. Membranes were incubated with primary antibodies of Runx2 (1:1000, Affinity, China), OCN (1:1000, Affinity, China), OPN (1:1000, Affinity, China) and COL I (1:500, Affinity, China) overnight at 4 °C, followed by the incubation of HRP-conjugated secondary antibodies. The proteins were visualised with an enhanced chemiluminescence (ECL)-chemiluminescent kit (Thermo Fisher Scientific, USA), and the images were acquired using Scion image software.

### Subcutaneous implantation of silk fibroin/magnesium composite scaffolds

Twelve 6-week-old male Sprague-Dawley rats weighing 220–230 g were used for in vivo subcutaneous implantation. All procedures were approved by the animal care and use committee of Beijing Jishuitan Hospital regarding the Guiding Principles for the care and use of experimental animals (Number: 2022-12-03). The scaffolds with a cylindrical shape (1 cm diameter, 2 mm height) were sterilised using cobalt-60 (25 kGy) radiation for 2 h. All the rats were anaesthetised with 0.5% pentobarbital sodium. Two small midline incisions were made on the dorsum of each rat, and the scaffolds were introduced into bilateral subcutaneous pockets created by blunt dissection. After 7, 14 and 28 days, the rats were sacrificed, and the implanted scaffolds with the surrounding tissue were removed. The samples were fixed and processed for histology studies.

### Rat skull defect model of silk fibroin/magnesium composite scaffolds

Twenty-four 6-week-old male Sprague-Dawley rats weighing 220–230 g were used to assess the skull regeneration ability of the scaffolds. All procedures of the animal experiments were approved by the Beijing Jishuitan Hospital Animal Care and Use Committee (Number: 2022-12-03). Blank group (defects only), control group (defects received SF scaffolds) and experimental group (defects received SF/1MgO scaffolds). After anaesthesia and routine preparation, two critical-size defects with a diameter of 4 mm were created in each rat. The scaffolds with a cylindrical shape (4 mm diameter, 1 mm height) were sterilised using cobalt-60 (25 kGy) radiation for 2 h. The scaffolds were implanted and the defective tissues were supported. The physical examination was monitored daily throughout the experimental period after the operation. At 4 and 8 weeks post-surgery, the rats were sacrificed by injecting an overdose of sodium pentobarbital, and the implanted scaffolds with the surrounding tissue were removed. The samples were fixed and processed for further studies.

### Micro-CT evaluation

All samples were scanned by a micro-CT machine (SCANCO Medical, MicroCT 100, Switzerland) to determine the 3D structure of the bone and the growth of newly grown bone tissue. The scanning parameters were 1 mm aluminium, 9 μm resolution, 70 kV voltage and 120 μA current. After scanning, 3D reconstruction of the bone was realised using a CT analyser, and the percentages of bone volume to total bone volume (BV/TV) and the local volumatric BMD were determined.

### Histology, immunohistochemistry and immunofluorescence staining

For in vivo histocompatibility, the sliced sections of subcutaneous implantation were subjected to H&E staining and Masson's trichrome staining to assess the tissue integration and FBR, including cell infiltration, the number of FBGC and collagenous fibrotic capsules formation of the scaffolds. For ectopic bone formation, the sliced sections of subcutaneous implantation were incubated with anti-OCN (1:100, Affinity, China) and anti-CD31 (1:200, Affinity, China) primary antibodies, followed by treatment with HRP-conjugated or Alexa Fluor

594-labelled secondary antibodies according to a standard protocol. For histological analysis of skull reconstruction, H&E and Masson's trichrome staining were performed. To further evaluate bone formation and angiogenesis, the OCN (1:100, Affinity, China), bone morphogenetic protein type 2 (BMP-2) (1:100, Affinity, China), Runx2 (1:100, Affinity, China), VEGF (1:100, Affinity, China) and CD31 antibodies (1:200, Affinity, China) were detected via immunofluorescence or immunohistochemistry staining. Images from stained sections were obtained using a digital slide scanner (3DHISTECH, Hungary), and image analysis was performed using IPP 6.0 to quantify the expression.

### Statistical analysis

At least three times each experiment was repeated independently with similar results and the results could be reproduced according to this method. Statistical analyses were carried out using SPSS version 20.0. The results are expressed as mean ± SD (standard deviation). The two-tailed unpaired Student's $t$ test was used to compare two groups, and a one-way ANOVA with Tukey's multiple comparisons was used to compare more than two groups. Statistical differences were shown with three significance levels. ns: $P > 0.05$, $*P < 0.05$, $**P < 0.01$, and $***P < 0.001$.

### Reporting summary

Further information on research design is available in the Nature Portfolio Reporting Summary linked to this article.

## Data availability

All relevant data that support the findings of this study are available within the article and Supplementary Information. All data are available from the corresponding authors upon request. Source data are provided with this paper.

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

## Acknowledgements

This work was financially supported by National Natural Science Foundation of China 32201114 (Z.M.), 82002303 (J.S.) and 82030067 (B.Y.), the Guangdong Basic and Applied Basic Research Foundation 2022A1515220079 (Z.M.), 2022A1515011536 (J.S.), 2021A1515220093 (J.S.), 2021A1515220086 (C.Y.), Shenzhen Science and Technology Program JCYJ20230807095116032 (X.B.), Beijing Municipal Health Commission BMHC-2021-6 (X.S.) and BJRITO-RDP-2023 (X.S.), Shenzhen Science and Technology Innovation Committee Project SGDX20220530111405038 (J.S.), GJHZ20210705142543019 (C.Y.).

## Author contributions

Z.M. and X.B. contributed equally to the work. Designed the research: Z.M., X.B., Y.Z. and X.S. Materials SF/MgO synthesized: Z.M. and X.B. In vitro and in vivo experiments: Z. M., X.B., C.Y., L.C., J.S., Y.H., Z.W. and H.Q. Prepared figures: M.Z. Manuscript writing and editing: Z.M. and B.X. Revised the manuscript: Z.M., X.B., B.Y., Y.Z. and J.G.

## Competing interests

The authors declare no competing interests.

## Additional information

[1]Department of Spine Surgery,Shenzhen Engineering Laboratory of Orthopaedic Regenerative Technologies, Peking University Shenzhen Hospital, Shenzhen Peking University-The Hong Kong University of Science and Technology Medical Center, Shenzhen, Guangdong province, China. [2]School of Materials Science and Engineering, Peking University, Beijing 100871, China. [3]Beijing Research Institute of Orthopedics and Traumatology, Beijing Jishuitan Hospital, Capital Medical University, Beijing 100035, China. [4]Technical University of Munich, TUM School of Life Sciences, Maximus-von-Imhof-Forum 2, D-85354 Freising, Germany. [5]International Research Center for Advanced Structural and Biomaterials, School of Materials Science & Engineering, Beihang University, Beijing 100191, China. [6]These authors contributed equally: Zhinan Mao, Xuewei Bi. ✉e-mail: shuxiong@jst-hosp.com.cn; hpyubinsheng@hotmail.com; yfzheng@pku.edu.cn

