## [Peer Review File · Nature Communications]

Reviewers' Comments:

Reviewer #1:

Remarks to the Author:

The authors report silk fibroin/magnesium (SF/MgO) [0, 10, and 30 wt% MgO] composite scaffold (formed by cryogelation) that may be readily trimmed and exhibit blood-/water-responsive shape memory behavior for the regenerative healing of irregularly shaped, critical-sized bone defects. While shape memory polymer (SMP) scaffolds have been previously proposed in the literature, including based on SF, the authors look to patient blood at the surgical site as the "trigger" for fitting. The combination of SF ("-" charged) and MgO ("+" charged) provides electrostatic crosslinking, in addition to the covalent and physical crosslinking, which is interesting to leverage. MgO also is leveraged for bioactivity. Overall, this is a very comprehensive study that covers biomaterial design, as well as in vitro and in vivo analyses. Schematics are nicely done in terms of capturing scaffold fabrication, and presentation of the data. A few points are made below to afford more context for the work, as well as clarification of some experimental protocols.

Introduction

Page 3: "Therefore, it is worth looking forward to developing a structurally stable and shape memory porous scaffold that could adapt to the irregular-shaped bone defect to achieve minimally invasive treatment." I disagree with the argument for "minimally invasive treatment". Rather the utility of shape memory polymer (SMP) scaffolds is achievement of conformal fitting for enhanced osseointegration and overall regeneration.

Page 3: "Calcium phosphate bone cement (CPC) is a popular injectable biomaterial owing to its well biocompatibility, osteoconductivity and its fitting ability,...". Instead of "fitting ability", I suggest "ability to fill irregularly shaped defects prior to cure, although this is associated with exotherms and shrinkage" or something to that effect.

"Shape memory polymers (SMPs), a kind of emerging intelligent biomaterials, have also brought great hope to patients with bone defects." Additional references are needed. For instance, Grunlan and co-workers have reported the use of thermoresponsive self-fitting polyester scaffolds for the last decade [e.g., Zhang, D. et al., "A bioactive "self-fitting" shape memory polymer (SMP) scaffold with potential to treat craniomaxillofacial (CMF) bone defects," *Acta Biomaterialia*, 2014, 10, 4597-4605; Woodard, L.N., et al. "Porous poly(ϵ -caprolactone)-poly(L-lactic acid) semi-interpenetrating networks as superior, defect-specific scaffolds with potential for cranial bone defect repair," *Biomacromolecules*, 2017, 18, 4075-4083; Pfau, M.R.; et al., "Shape memory polymer (SMP) bone scaffolds with improved self-fitting properties," *J. Mater. Chem. B*, 2021, 9, 3286-3837.; Pfau, M.R., et al. "Evaluation of self-fitting, shape memory polymer scaffolds in a rabbit calvarial defect model," *Acta Biomaterialia*, 2021, 136, 233-242.] and also P. Rychter, et al. "Scaffolds with shape memory behavior for the treatment of large bone defects," *J. Biomed. Mater. Res. A*, 2015, 103, 3503-3515.

"However, traditional silk fibroin materials do not have shape memory effect, and the structure will be permanently damaged under external force, resulting in a tight fit with bone defects after implantation." This sentence is contradictory and does not make sense.

Overall, the authors go back-and-forth between "water-responsive" and "blood-responsive". The authors should highlight water-responsive shape memory systems in the literature.

Even with blood-responsivity, there is a potential "thermoresponsive" aspect for deployment at body temperature. [e.g., J. Shang, et al, *Polym. Chem.*, 2019, 10, 1036-1055. M.R. Pfau, et al. *J. Mater. Chem. B*, 2021, 9, 4287-4297.]

Results

Page 6 and Figure 1: Suggest putting scaffold composition notation ["SF", "SF-1nMgO", and "SF-3nMgO"] in both text of page 6 and Figure 1 to improve manuscript clarity.

Page 5: "After 24 h of cryogelation, scanning electron microscopy (SEM) images confirmed that the

three scaffolds had continuous porous structures with pore sizes of $\sim 100 \mu\text{m}$ (Fig. 2a, and Supplementary Fig. 2, 3)." Per page 33, no information is given in regard to specimen preparation for SEM. If this involves freeze-drying, the authors must acknowledge that this process can alter/introduce artifacts. Thus, a literal interpretation of the SEM images should be tempered.

Water content is not reported; this will impact mechanical properties, and responsiveness to blood/water. Mechanical properties are only related to MgO content, but accompanied reduced water content should be considered.

Compressive moduli of the scaffolds are $< 0.3 \text{ MPa}$. The authors should comment on the limitation of this in the context of bone healing (e.g., in situation where load is applied).

Figure 4: What was the temperature of the shape recovery tests (i.e., was water and blood temperature RT or body temperature)? This should be added to the Experimental Section (e.g., specimen size, test temperature, etc).

Reviewer #2:

Remarks to the Author:

This is an interesting manuscript. But the experimental design and data analysis need to be improved. Following questions need to be answered.

#1

How to ensure that the scaffolds is sterile before being implanted into the body?

#2

In line 255, the author believes that "SF/MgO composite scaffolds had good blood-responsive shape memory effect". In my opinion, the scaffolds has good shape memory ability, but neither the Fig3-4 nor the supplementary video can prove that water or blood is the reason why the scaffolds returns to its original shape. Because in the video, the scaffolds are put into a bone mold or in water under external extrusion, and in a dry environment, the scaffolds also shows a great degree of shape recovery.

Please provide data on the volume change of the scaffolds in the air within 15 seconds after the water is squeezed out without interference from external forces.

#3

Please explain how water or blood speeds up the scaffold's rebound?

#4

The author believes that the scaffolds can be used for Irregular Bone Regeneration, but only a critical-sized skull defects model was designed in the experiment, and an irregular-shaped bone defect model was not established on this basis.

Reviewer #3:

Remarks to the Author:

The authors reported silk fibroin/magnesium composite scaffolds with improved mechanical properties and shape-memory combining chemical and physical crosslinking for irregular bone regeneration. This scaffold could promote cell proliferation, adhesion, migration of osteoblasts, and osteogenic differentiation of BMSCs in vitro and could promote in situ bone regeneration of cranial defect, which are interesting. However, the statement "excellent mechanical performance" is not convinced and whether the degradability of such composite scaffold is suitable for bone regeneration is not clearly supported, I think. Previous studies have proved that magnesium could also regulate the cell functions including proliferation, adhesion, and migration, as well as promote neovascularization. I cannot see any milestones achieved to meet the standard of Nature Communications, I believe. Some comments are as follows:

1. The mechanical properties of the SF/MgO composite scaffolds were shown in Fig. 3, around few

hundred kPa, which are not high for bone implants and bone tissue. Please comment it.

2. Is it strange that "The addition of MgO particles reduced the degradation rate of SF scaffolds " and "the degradation rate of the three scaffolds followed the order: SF-3nMgO > SF-1nMgO > SF (Fig. 2f)" and "However, the degradation rate of the three groups is not different. "on lines 195-200? Whether the degradability of such composite scaffold is suitable for bone regeneration is not clearly supported.

3. How about the data of the in vivo bone regeneration after longer time such as 12-weeks, because bone regeneration needs longer time.

4. Residual crosslinker EGDE was not removed? How about the toxicity?

5. On Line 149, "EGDE"? Line 192, what are "SF-1nMgO, and SF-3nMgO"? "Silk Fibroin/Magnesium" on lines 275-276, why are the uppercase letters used here? Some errors such as line 390 "8 weeks after surgery, The 3D reconstruction image as shown in Fig. 9b."

Response to Reviewer #1:

“The authors report silk fibroin/magnesium (SF/MgO) [0, 10, and 30 wt% MgO] composite scaffold (formed by cryogelation) that may be readily trimmed and exhibit blood-/water-responsive shape memory behavior for the regenerative healing of irregularly shaped, critical-sized bone defects. While shape memory polymer (SMP) scaffolds have been previously proposed in the literature, including based on SF, the authors look to patient blood at the surgical site as the "trigger" for fitting. The combination of SF ("-“ charged) and Mg ("+" charged) provides electrostatic crosslinking, in addition to the covalent and physical crosslinking, which is interesting to leverage. MgO also is leveraged for bioactivity. Overall, this is a very comprehensive study that covers biomaterial design, as well as in vitro and in vivo analyses. Schematics are nicely done in terms of capturing scaffold fabrication, and presentation of the data. A few points are made below to afford more context for the work, as well as clarification of some experimental protocols.”

Response: Thanks for your affirmation and the nice comment on this work. The experimental section has been thoroughly checked and optimized.

Introduction

“Page 3: "Therefore, it is worth looking forward to developing a structurally stable and shape memory porous scaffold that could adapt to the irregular-shaped bone defect to achieve minimally invasive treatment." I disagree with the argument for "minimally invasive treatment". Rather the utility of shape memory polymer (MP) scaffolds is achievement of conformal fitting for enhanced osseointegration and overall regeneration.”

Response: Thanks for your valuable comment. I completely agree with Reviewer’s the statement “Rather the utility of shape memory polymer (MP) scaffolds is achievement of conformal fitting for enhanced osseointegration and overall regeneration.”

This is one of the important properties of the scaffolds. However, most bone defects are of irregular shape in clinical, more extreme cases, such as the diameter of the surface defect is smaller than internal defect. And, it is usually necessary to enlarge the bone defect into a regular one by open surgery, which increases the difficulty and cost of the operation, and brings severe trauma, pain and severe dysfunction to the patient. As a result, the recovery rate of the patient's wound is also slowed. However, the shape memory SF/MgO composite scaffolds can be processed into large blocks, which can be trimmed during the operation according to demand. The aim is to avoid enlarging the irregular bone defect into a regular bone defect. We anticipate that these bioactive shape memory porous scaffolds have great clinical application potential in minimally invasive clinical-size irregular bone regeneration.

“Page 3: "Calcium phosphate bone cement (CPC) is a popular injectable biomaterial owing to well biocompatibility, osteoconductivity and its fitting ability,... Instead of "fitting ability", I suggest "ability to fill irregularly shaped defects prior to cure, although this is associated with exotherms and shrinkage" or something to that effect.”

Response: Thanks for your valuable comment. The detailed content has been revised in the manuscript with red font. “Calcium phosphate bone cement (CPC) is a popular injectable biomaterial in orthopaedic surgeries because of the good biocompatibility, osteoconductivity and the ability to fill the irregularly shaped defects, and it has been used for various micro-invasive patterns (such as vertebroplasty and arthroscopic bone repairing).”

“Shape memory polymers (SMPs), a kind of emerging intelligent biomaterials, have also brought great hope to patients with bone defects. " Additional references are needed.”

Response: Thanks for your valuable comment. The references have been added accordingly.

“However, traditional silk fibroin materials do not have shape memory effect, and the structure will be permanently damaged under external force, resulting in a tight fit with bone defects after implantation." This sentence is contradictory and does not make sense.”

Response: Thanks for your valuable comment. The detailed content has been revised in the manuscript with red font. “Traditional SF materials do not have a shape-memory effect, and their structure is permanently damaged under external force, resulting in a poor integration fit with bone defects after implantation.”

“Overall, the authors go back-and-forth between "water-responsive" and "blood-responsive". The authors should highlight water-responsive shape memory systems in the literature.”

Response: Thanks for your valuable comment. As suggested by the reviewer, we have unified the two expressions as **water-responsive** shape memory in the manuscript.

“Even with blood-responsivity, there is a potential "thermoreponsive" aspect for deployment at body temperature. [e.g., J. Shang, et al, Polym. Chem., 2019, 10, 1036-1055. M.R. Pfau, et al. J. Mater. Chem. B, 2021,9,4287-4297]”

Response: Thanks for your valuable comment. Around 60 °C, Silk fibroin displays an endothermic relaxation, usually related to an increase of the amorphous domain motion or protein-water bonding rupture, which induces an increase in the molecular motions and in the plasticity of SF materials [1-3]. We also verified the thermal response characteristics of the SF-1nMgO scaffolds. The results showed that the deformed dry SF-1nMgO scaffolds cannot recover the original shape at 37 °C [Supplementary Video 7]. Silk fibroin has a sophisticated hierarchical structure consisting of less ordered hydrophilic (amorphous domains) and crystallizable hydrophobic blocks (β -crystals domains) [4, 5]. Hydrophilic blocks provide solubility in water and are responsible for fibroin elasticity and toughness. Therefore, according to the performance characteristics of the silk fibroin, the responsiveness of the scaffolds at room temperature and body temperature is determined by less ordered hydrophilic blocks.

- [1] Hu, X., Kaplan, D., Cebe, P. Effect of water on the thermal properties of silk fibroin. *Thermochim. Acta* **461**, 137-144 (2007).
- [2] Marelli, B. N. et al. Programming function into mechanical forms by directed assembly of silk bulk materials. *Proc. Natl. Acad. Sci. U. S. A.* **114**, 451-456 (2017).
- [3] Tulachan, B. et al. Electricity from the Silk Cocoon Membrane, *Sci. Rep.* **4**, 5434 (2014).
- [4] Atkins, E. Silk's secrets. *Nature* **424**, 1010-1010 (2003).
- [5] Sahoo, J.K., Hasturk, O., Falcucci, T., Kaplan, D.L., Silk chemistry and biomedical material designs. *Nat. Rev. Chem.* DOI:10.1038/s41570-023-00486-x (2023).

Results

“Page 6 and Figure 1: Suggest putting scaffold composition notation ["SF", "SF-1nMgO", and "SF-3nMgO"] in both text of page 6 and Figure 1 to improve manuscript clarity.”

Response: Thanks for your valuable comment and advice. The detailed content has been added accordingly and in the revised manuscript with red font.

“According to the mass ratio of the nano-MgO particles to the solute mass of the SF solution at 0 wt%, 10 wt% and 30 wt%, named SF, SF-1nMgO and SF-3nMgO, respectively.”

“Page 5: "After 24 h of cryogelation, scanning electron microscopy (SEM) images confirmed that the three scaffolds had continuous porous structures with pore sizes of ~100 um (Fig. 2a, and Supplementary Fig. 2, 3)." Per page 33, no information is given in regard to specimen preparation for SEM. If this involves freeze-drying, the authors must acknowledge that this process can alter/introduce artifacts. Thus, a literal interpretation of the SEM images should be tempered.”

Response: Thanks for your valuable comment and advice. The specimen preparation for SEM has been supplemented and illustrated in the Experimental Methods section. The scaffolds do need to be lyophilized prior to SEM testing. In this process, for structurally stable scaffolds, our team found that the effect on the pore size of the scaffolds was still very limited during freeze-drying.

“The pore size and micro-morphology of SF/MgO scaffolds were examined by scanning electron microscope with energy dispersive spectrometer (SEM-EDS, S-4800, Hitachi, Japan) at an accelerating voltage of 10 kV. The cryogelled SF/MgO composite scaffolds were first frozen at -20 °C for 12 h and vacuum-dried at -50 °C for 24 h. The dry SF/MgO scaffolds were sputter-coated with gold for SEM test.”

“Water content is not reported; this will impact mechanical properties, and responsiveness to blood/water. Mechanical properties are only related to MgO content, but accompanied reduced water content should be considered.”

Response: Thanks for your valuable comment. As stated by the reviewer, the water content of the scaffolds has a very strong influence on its mechanical properties and water responsiveness. The test conditions have been supplemented and illustrated in the

Experimental Methods section. To equilibrate, the scaffold samples were fully saturated in PBS prior to mechanical testing. Therefore, there is a direct relationship between the mechanical properties of scaffolds and MgO content.

“The mechanical retention of samples was incubated at 37°C in SBF solution and removed at indicated time points (0, 14 and 28 days). On day 0, all samples were adequately hydrated in PBS for 12 h before testing. The compression test was performed at a strain rate of 30%/min. Young’s modulus of scaffolds was calculated from the initial linear strain range of the curve obtained from the stress-strain. Cyclic compression testing of the SF-1nMgO scaffolds was performed with a maximum strain of 30% for five cycles. Dynamic cyclic compression testing of the SF-1nMgO scaffolds was performed at a strain rate of -100%/min to -22% strain and 100%/min to 2% strain for 1,000 cycles.”

“Compressive moduli of the scaffolds are <0.3 MPa. The authors should comment on the limitation of this in the context of bone healing (e.g., in situation where load is applied).”

Response: Thanks for your valuable comment and advice. The scaffolds in this study are mainly applied to bone repair at non-load-bearing sites. Using its elastic, adaptive and tailorable properties, it can achieve minimally invasive treatment of irregular bone defects. As noted, the scaffolds do indeed have limitations, which do not meet the bone repair at load-bearing sites. We have therefore revised the relevant claims in the Discussion.

“There are still limitations to our study. The compression modulus of the SF/MgO composite scaffolds is between 0.1 and 0.3 MPa, which is suitable for non-load-bearing bone repair; an irregular-shaped bone defect model should be established that could fully demonstrate and reflect the performance of the SF/MgO composite scaffolds. The time spent studying animal models needs to be extended to evaluate the long-term bone repair effects of SF/MgO composite scaffolds. In future studies, a large experimental animal model should be established to further verify the concept of minimally invasive treatment of irregular bone defects and to explore the related molecular mechanism of osteogenesis.”

“Figure 4: What was the temperature of the shape recovery tests (i.e., was water and blood temperature RT or body temperature)? This should be added to the Experimental Section (e.g., specimen size, test temperature, etc).”

Response: Thanks for your valuable comment and advice. The shape recovery test of the scaffolds was performed under room temperature conditions. The experimental section has been thoroughly checked and optimized.

“The SF/MgO scaffolds were fully hydrated in PBS solution and subsequently subjected to bending, compression and torsion deformation, followed by exposure to room-temperature water to evaluate their ability to restore their original shape. Multiple scaffolds were completely compressed together and subsequently exposed to room-temperature water and blood to record the rate and extent of recovery of the scaffolds

to their original shape in different media environments. In addition, the compressed wet SF-1nMgO scaffolds were placed at a weight of 50 g. The scaffolds were then incubated in room-temperature water and the images were recorded. Furthermore, the potential of SF-1nMgO scaffolds to be trimmed during clinical surgery for personalised implantation was explored. The bulk SF-1nMgO scaffolds were prepared and then roughly and precisely trimmed according to the shape of the bone defect. Finally, the SF-1nMgO scaffolds were wetted and compressed to achieve the implantation of small pore-size irregular bone defects.”

Response to Reviewer #2:

This is an interesting manuscript. But the experimental design and data analysis need to be improved. Following questions need to be answered.

Response: Thank you very much for your affirmation of this work. We have tried our best to address these concerns and improve the clarity of this work. As suggested, the experimental design and data analysis have been comprehensively improved in this manuscript.

#1. How to ensure that the scaffolds is sterile before being implanted into the body?

Response: Thanks for your valuable comment. The detailed content has been added to the experimental method. “The scaffolds were sterilized by cobalt-60 (25kGy) radiation for 2 hours and then implanted.”

#2. In line 255, the author believes that "SF/Mg composite scaffolds had good blood-responsive shape memory effect". In my opinion, the scaffolds have good shape memory ability, but neither the Fig3-4 nor the supplementary video can prove that water or blood is the reason why the scaffolds return to its original shape. Because in the video, the scaffolds are put into a bone mold or in water under external extrusion, and in a dry environment, the scaffolds also show a great degree of shape recovery.

Please provide data on the volume change of the scaffolds in the air within 15 seconds after the water is squeezed out without interference from external forces.

Response: Thanks for your valuable comment. As the reviewer states, the scaffolds have good shape memory ability and show a great degree of shape recovery. We made a prominent statement and a critical experiment on this issue. The details are as follows:

Firstly, scaffolds in dry state will undergo permanent deformation after being subjected to external forces and do not have shape recovery ability [Fig. 1]. However, when the deformed dry scaffolds were in contact with water/blood, they would gradually restore their original shape. At the same time, we also admit that scaffolds in wet state or with a certain water content will be deformed under the action of external force, and scaffolds have a certain shape recovery ability after external force unloading. However,

it should be emphasized that the scaffolds do not have the ability to completely restore the original shape without contact with water/blood. Therefore, either dry or wet state deformed scaffolds have the ability to fully recover the original shape only when they are in contact with water or blood [Fig. S5 and Supplementary Video 6]. This is also the driving force for the scaffolds to achieve complete rebound and close contact with irregular tissues after being compressed and implanted into the body. Finally, we also verified the volume change value of the scaffolds within 30 seconds after the water is squeezed out without interference from external forces. After our multiple estimates, the volume of the scaffolds after rebound was ~80 % of the original volume.

“Fig. 4b, Supplementary Fig. 5 and Supplementary Videos 4-7 provided a robust demonstration that deformed SF/MgO scaffolds have the ability to fully recover the original shape only when in contact with water or blood.”

Fig. S5 Schematic illustration of water-responsive shape-memory effect SF-1nMgO scaffolds. The SF-1nMgO scaffolds in the dry state will be permanently deformed after being subjected to external force, and has no shape recovery ability. However, the deformed dry scaffolds were in contact with water/blood, they would gradually restore their original shape.

#3. Please explain how water or blood speeds up the scaffold's rebound?

Response: Thanks for your valuable comment. There are three main reasons why water or blood is able to accelerate scaffolds rebound:

1. The hydrophilic/hydrophobicity of the material itself has a direct impact on the water-/blood- responsive capacity of the scaffolds.

(i) Silk fibroin has a sophisticated hierarchical structure consisting of less ordered hydrophilic (amorphous domains) and crystallizable hydrophobic blocks (β -crystals domains) [4, 5]. Hydrophilic blocks provide solubility in water and are responsible for fibroin elasticity and toughness, while hydrophobic blocks form intermolecular β -sheet structures leading to the insolubility and high strength of fibroin.

2. The scaffolds have excellent water-/blood-responsive shape-memory effect, which is inseparable from excellent mechanical stability and structural retention.

(i) The mechanical properties of silk scaffolds are dominated by its cross-linked configuration, and SF amorphous domains firstly support the external forces [6]. The scaffolds in this study are multiple cross-linking network structural materials integrating covalent cross-linking, physical cross-linking, and electrostatic interactions. After the scaffolds are subjected to compression, the SF amorphous domains will be deformed but the structure will not be damaged. In this process, internal stress will be generated in the scaffolds to restore the original state [7]. This reason is also the basis and prerequisite for scaffolds to have water or blood responsiveness.

(ii) β -crystals domain arrangements, size, and orientation also influence the mechanical properties. β -crystals confined to a few nanometers can achieve higher strength, stiffness, and toughness than larger β -crystallites. The reported dimensions of β -crystallites for optimal mechanical properties are between 1 and 4 nm [8]. It is worth noting that our previous research work has demonstrated that β -crystals of SF scaffolds prepared by our optimized cryogelation technology are confined to 1-2 nanometers [9]. This is another reason why the scaffolds have blood-responsive shape-memory effect. The relevant explanations and analyses have been added and in the revised manuscript with red font.

“The main explanation for this phenomenon falls into three categories: (i) SF has a sophisticated hierarchical structure consisting of less ordered hydrophilic (amorphous domains) and crystallisable hydrophobic blocks (β -crystals domains). Hydrophilic blocks provide solubility in water and are responsible for fibroin elasticity and toughness; (ii) the SF/MgO composite scaffolds are subjected to compression, and the SF amorphous domains deform first, but the structure remains intact. During this process, internal stress is generated in the scaffolds to restore the original state; and (iii) domain arrangements, size and orientation of the β -crystals also influence the mechanical properties. β -crystals of SF scaffolds prepared by our optimised cryogelation technology are confined to 1-2 nanometers, which can achieve higher strength, stiffness and toughness.”

[4] Atkins, E. Silk's secrets. *Nature* **424**, 1010-1010 (2003).

[5] Sahoo, J.K., Hasturk, O., Falcucci, T., Kaplan, D.L., Silk chemistry and biomedical material designs. *Nat. Rev. Chem.* DOI:10.1038/s41570-023-00486-x (2023).

[6] Reizabal, A., Costa, C.M., Pérez-Alvarez, L., Vilas-Vilela, J.L. & Lanceros-Méndez, S. Silk Fibroin as Sustainable Advanced Material: Material Properties and Characteristics, Processing, and Applications. *Adv. Funct. Mater.* **33**, 2023.

[7] Ak, F., Oztoprak, Z., Karakutuk, I., Okay, O. Macroporous Silk Fibroin Cryogels. *Biomacromolecules* **14**, 719-727 (2013).

[8] Koh, L.D. et al. Structures, mechanical properties and applications of silk fibroin materials. *Prog. Polym. Sci.* **46**, 86-110 (2015).

[9] Mao, Z.N. et al. Controlled cryogelation and catalytic cross-linking yields highly elastic and robust silk fibroin scaffolds. *ACS Biomater. Sci. Eng.* **6**, 4512-4522 (2020).

#4. The author believes that the scaffolds can be used for Irregular Bone Regeneration, but only a critical-sized skull defects model was designed in the experiment, and an irregular-shaped bone

defect model was not established on this basis.

Response: Thanks for your valuable comment. We regret that the irregular-shaped bone defect model was not established in this work. In fact, the establishment of irregular bone defect model is still in the development stage, and there is no unified standard. And more scholars are needed to participate in the discussion of the establishment of irregular bone defect model. Nevertheless, the previous materials research on the repair and regeneration of irregularly shaped bone defects were still mainly based on critical-sized skull defects model [10-13]. In the future, we will revise the animal defect model for the evaluation of the effects of irregular-shaped bone repair and regeneration.

[10] Liu, Q. et al. Highly Malleable Personalized Prostheses with Hierarchical Microstructure Boost the Long-Term Osteointegration in Irregular Craniofacial Reconstruction. *Adv. Funct. Mater.* DOI:10.1002/adfm.202304308 (2023).

[11] Li, Y.M. et al. Injectable Biomimetic Hydrogel Guided Functional Bone Regeneration by Adapting Material Degradation to Tissue Healing. *Adv. Funct. Mater.* DOI:10.1002/adfm.202213047 (2023).

[12] Zou, Y.P. et al. Precipitation-Based Silk Fibroin Fast Gelling, Highly Adhesive, and Magnetic Nanocomposite Hydrogel for Repair of Irregular Bone Defects. *Adv. Funct. Mater.* **33**, (2023).

[13] Lu, G.G. et al. An instantly fixable and self-adaptive scaffold for skull regeneration by autologous stem cell recruitment and angiogenesis. *Nat. Commun.* **13**, (2022).

Response to Reviewer #3:

The authors reported silk fibroin/magnesium composite scaffolds with improved mechanical properties and shape-memory combining chemical and physical crosslinking for irregular bone regeneration. This scaffold could promote cell proliferation, adhesion, migration of osteoblasts, and osteogenic differentiation of BMSCs in vitro and could promote in situ bone regeneration of cranial defect, which are interesting. However, the statement "excellent mechanical performance" is not convinced and whether the degradability of such composite scaffold is suitable for bone regeneration is not clearly supported, I think. Previous studies have proved that magnesium could also regulate the cell functions including proliferation, adhesion, and migration, as well as promote neovascularization. I cannot see any milestones achieved to meet the standard of Nature Communications; I believe. Some comments are as follows:

Response: Thank you very much for your interest and the nice comment on this work. The statement "excellent mechanical performance" has been revised to "excellent mechanical stability". According to the reviewer's comments, whether the degradation rate of the scaffolds is suitable for bone regeneration, we have therefore revised the relevant claims in the discussion. As the reviewer's comments, previous studies have proved the biological functions of Mg²⁺. Our group have also made great efforts on the role of Mg metal/Mg²⁺ in promoting osteogenesis, and revealed the mechanism of Mg²⁺ in promoting bone regeneration [14-16]. Meanwhile, our previous study has in-depth research on the relationship between the molecular structure of silk fibroin and its

physicochemical and biological properties [17, 18]. At the same time, the regeneration of critical-sized bone defects, especially for the ones with irregular shapes, remains a clinical challenge. Therefore, based on our previous extensive studies, this work creatively constructed a water-responsive shape memory effect silk fibroin/ magnesium (SF/MgO) composite scaffold, which was able to easily trimmed, compacted and implanted into the body, and fully recovered original shape in contact with water or blood, and it could provide mechanical support for tight-contact with surrounding tissues. The addition of MgO particles in SF scaffolds obviously enhanced the biological activity; on the other hand, it brought more possibilities for the regulation of the degradation rate of scaffolds for bone repair. The SF/MgO composite scaffold basically meets the requirements of minimally invasive implantation of irregular bone defects.

“In vivo cranial bone repair generally took over 8 weeks, indicating that the degradation duration of scaffolds should meet the requirement of the healing time. At 8 weeks after SF-1nMgO scaffold implantation, the histological staining results showed partial degradation of the SF-1nMgO scaffolds, accompanied by in situ regeneration of new bone tissue inside the scaffolds. Therefore, a prolonged implantation time of the SF-1nMgO scaffolds meant the remaining MgO inside the degrading scaffolds would gradually hydrolyse and release Mg^{2+} to promote bone repair. By contrast, SF scaffolds were largely degraded after 8 weeks of implantation, but very limited new bone formation was seen. Therefore, while the addition of MgO particles to SF scaffolds enhanced the biological activity, it also brought more possibilities for regulating the degradation rate of scaffolds for bone repair.”

- [14] Shen, J. et al. Stepwise 3D-spatio-temporal magnesium cationic niche: Nanocomposite scaffold mediated microenvironment for modulating intramembranous ossification. *Bioact. Mater.* **6**, 503-519 (2021).
- [15] Wong, H.M. et al. Low-modulus Mg/PCL hybrid bone substitute for osteoporotic fracture fixation. *Biomaterials* **34**, 7016-7032 (2013).
- [16] Zhang, Y.F. et al. Implant-derived magnesium induces local neuronal production of CGRP to improve bone-fracture healing in rats. *Nat. Med.* **22**, 1160-1169 (2016).
- [17] Mao, Z.N. et al. Controlled Cryogelation and Catalytic Cross-Linking Yields Highly Elastic and Robust Silk Fibroin Scaffolds. *ACS Biomater. Sci. Eng.* **6**, 4512-4522 (2020).
- [18] Mao, Z.N. et al. The relationship between crosslinking structure and silk fibroin scaffold performance for soft tissue engineering. *Int. J. Biol. Macromol.* **182**, 1268-1277 (2021).

1. The mechanical properties of the SF/Mg composite scaffolds were shown in Fig. 3, around few hundred kPa, which are not high for bone implants and bone tissue. Please comment it.

Response: Thanks for your valuable comment. The scaffolds in this study are mainly applied to bone repair at non-load-bearing sites. Using the elastic, adaptive and tailorable properties of SF/MgO scaffolds, it can achieve minimally invasive treatment of irregular bone defects. Ideally, biomaterials with excellent osteoinductivity, suitable porosity, and biodegradability are preferred for non-load-bearing bone repair. More

importantly, the degradation rate of materials should match the bone regeneration rate, providing sufficient scaffold-derived support for cell growth and differentiation, meanwhile degrading to provide space for new bone growth [11, 19]. The detailed content has been added accordingly and in the revised manuscript with red font. “The elastic modulus was between 0.1 and 0.3 MPa, which was suitable for non-load-bearing bone repair”.

[11] Li, Y.M. et al. Injectable Biomimetic Hydrogel Guided Functional Bone Regeneration by Adapting Material Degradation to Tissue Healing. *Adv. Funct. Mater.* DOI:10.1002/adfm.202213047 (2023).

[19] Lee, S.S. et al. Sequential growth factor releasing double cryogel system for enhanced bone regeneration. *Biomaterials* **257**, (2020).

2. Is it strange that "The addition of Mg particles reduced the degradation rate of SF scaffolds and "the degradation rate of the three scaffolds followed the order: SF-3nMgO > SF-1nMgO > SF (Fig. 2f)" and "However, the degradation rate of the three groups is not different. "on lines 195-200? Whether the degradability of such composite scaffold is suitable for bone regeneration is not clearly supported.

Response: Thanks for your valuable comment. We apologize for the expression of "However, the degradation rate of the three groups is not different.". What we intend to express is that there is no significant difference in the conformational content of the three groups of materials. It must be admitted that the matching of the degradation rate of the materials with the regeneration rate of the new bone tissue is the core point of the design of the new materials. The *in vitro* degradation speed test of materials cannot provide a scientific criterion for the degradation trend *in vivo*, but it can be used to compare the degradation trend of materials under different parameters. According to the reviewer's comments, whether the degradation rate of the scaffolds is suitable for bone regeneration, we carry out a detailed and in-depth analysis in the discussion section.

“*In vivo* cranial bone regeneration generally took over 8 weeks, indicating degradation duration of scaffolds should meet the requirement of the tissue regenerate time. At 8 weeks after SF-1nMgO scaffolds implantation, the histological staining results showed partial degradation of the SF-1nMgO scaffolds, accompanied by *in situ* regeneration of new bone tissue inside the scaffolds. Therefore, if the implantation time of the SF-1nMgO scaffolds is prolonged; with the degradation of the remaining scaffolds, the remaining MgO inside the scaffolds will gradually hydrolyze and release Mg^{2+} to promote bone regeneration and repair. In contrast, SF scaffolds were largely degraded after 8 weeks of implantation, but there was limited new bone regeneration. Therefore, on the one hand, the addition of MgO particles in SF scaffolds enhances the biological activity of scaffolds; on the other hand, it brings more possibilities for the regulation of the degradation rate of scaffolds for bone repair.”

3. How about the data of the in vivo bone regeneration after longer time such as 12-weeks, because bone regeneration needs longer time.

Response: Thanks for your valuable comment and suggestion. We regret that the animal defect model was performed for 8 weeks. At 8 weeks after implantation, the micro-CT, histological and immunological staining analysis had showed that the SF/MgO group was significantly better than the blank control group and the SF group. Meanwhile, the data 8-week of SF/MgO group in this study was higher than the 8-week data and even close to the 12-week data of other studies through the horizontal comparison of bone volume fraction [19-21]. Nevertheless, previous studies on bone defect repair were conducted for 4 or 8 weeks [21-25]. At 8 weeks after scaffolds implantation, the histological staining results showed that partial degradation of the scaffolds occurred, accompanied by in situ regeneration of new bone tissue inside the scaffolds. It is well known that silk fibroin is an enzyme-responsive, fully degradable material. Therefore, if the implantation time of the scaffolds is prolonged; with the degradation of the remaining scaffolds, the remaining MgO inside the scaffolds will gradually hydrolyze and release Mg²⁺ to promote bone regeneration and repair. Therefore, we hypothesized that the bone repair effect of the scaffolds could be further improved by prolonging the implantation time. In the future, we will extend the animal model time for the evaluation of the effects of bone repair and regeneration.

- [19] Lee, S.S. et al. Sequential growth factor releasing double cryogel system for enhanced bone regeneration. *Biomaterials* **257**, (2020).
- [20] Zhang, Y.C. et al. 3D-printed NIR-responsive shape memory polyurethane/magnesium scaffolds with tight-contact for robust bone regeneration. *Bioact. Mater.* **16**, 218-231 (2022).
- [21] Qiao, W. et al. TRPM7 kinase-mediated immunomodulation in macrophage plays a central role in magnesium ion-induced bone regeneration. *Nat. Commun.* **12**, (2021).
- [22] Zhao, Z.Y. et al. Capturing Magnesium Ions Microfluidic Hydrogel Microspheres for Promoting Cancellous Bone Regeneration. *ACS Nano* **15**, 13041-13054 (2021).
- [23] Zhang, X.D. et al. Bioinspired Mild Photothermal Effect-Reinforced Multifunctional Fiber Scaffolds Promote Bone Regeneration. *ACS Nano* **17**, 6466-6479 (2023).
- [24] Liu, W.W. et al. In situ activation of flexible magnetoelectric membrane enhances bone defect repair. *Nat. Commun.* **14**, (2023).
- [25] Chen, S. et al. Local H₂ release remodels senescence microenvironment for improved repair of injured bone. *Nat. Commun.* **14**, (2023).

4. Residual crosslinker EGDE was not removed? How about the toxicity?

Response: Thanks for your valuable comment. In our previous work, attenuated total reflectance-fourier transform infrared spectroscopy (ATR-FTIR) results have demonstrated that residual EGDE can be completely removed from the scaffolds [9]. At the same time, the *in vitro* cytocompatibility of the SF scaffolds was assessed with bone marrow mesenchymal stem cells (BMSCs). The results showed that the cells had a good proliferation trend and cell morphology with the increase of culture time [9, 26].

- [9] Mao, Z.N. et al. Controlled cryogelation and catalytic cross-linking yields highly elastic and robust silk fibroin scaffolds. *ACS Biomater. Sci. Eng.* **6**, 4512-4522 (2020).
- [26] Mao, Z.N. et al. A Cell-Free Silk Fibroin Biomaterial Strategy Promotes In Situ Cartilage Regeneration Via Programmed Releases of Bioactive Molecules. *Adv. Healthc. Mater.* **12**, e2201588 (2023).

5. On Line 149 "EGDE"? Line 192, what are "SF-1nMgO, and SF-3nMgO"? "Silk Fibroin/Magnesium" on lines 275-276, why are the uppercase letters used here? Some errors such as line 390 "8 weeks after surgery, The 3D reconstruction image as shown in Fig. 9b.

Response: Thanks for your valuable comment.

In the revised manuscript, we have given specific explanations for "EGDE", "SF-1nMgO", and SF-3nMgO".

“ethylene glycol diglycidyl ether (EGDE)”

“According to the mass ratio of the nano-MgO particles to the solute mass of the SF solution at 0 wt%, 10 wt% and 30 wt%, named SF, SF-1nMgO and SF-3nMgO, respectively.”

"Silk Fibroin/Magnesium" on lines 275-276 has been revised to “silk fibroin/magnesium”.

“line 390 "8 weeks after surgery, The 3D reconstruction image as shown in Fig. 9b.” has been revised to “First, micro-CT analysis was used to assess bone regeneration at 4 and 8 weeks after surgery. Fig. 9b shows the 3D reconstruction images.”

The manuscript has been thoroughly checked and optimized, including grammar, experimental design, data analysis and some detail errors.

Response to Reviewer #1:

“The authors report silk fibroin/magnesium (SF/MgO) [0, 10, and 30 wt% MgO] composite scaffold (formed by cryogelation) that may be readily trimmed and exhibit blood-/water-responsive shape memory behavior for the regenerative healing of irregularly shaped, critical-sized bone defects. While shape memory polymer (SMP) scaffolds have been previously proposed in the literature, including based on SF, the authors look to patient blood at the surgical site as the "trigger" for fitting. The combination of SF ("- " charged) and Mg ("+" charged) provides electrostatic crosslinking, in addition to the covalent and physical crosslinking, which is interesting to leverage. MgO also is leveraged for bioactivity. Overall, this is a very comprehensive study that covers biomaterial design, as well as in vitro and in vivo analyses. Schematics are nicely done in terms of capturing scaffold fabrication, and presentation of the data. A few points are made below to afford more context for the work, as well as clarification of some experimental protocols.”

Response: Thanks for your affirmation and the nice comment on this work. The experimental section has been thoroughly checked and optimized.

Introduction

“Page 3: "Therefore, it is worth looking forward to developing a structurally stable and shape memory porous scaffold that could adapt to the irregular-shaped bone defect to achieve minimally invasive treatment." I disagree with the argument for "minimally invasive treatment". Rather the utility of shape memory polymer (MP) scaffolds is achievement of conformal fitting for enhanced osseointegration and overall regeneration.”

Response: Thanks for your valuable comment. I completely agree with Reviewer’s the statement “Rather the utility of shape memory polymer (MP) scaffolds is achievement of conformal fitting for enhanced osseointegration and overall regeneration.”

This is one of the important properties of the scaffolds. However, most bone defects are of irregular shape in clinical, more extreme cases, such as the diameter of the surface defect is smaller than internal defect. And, it is usually necessary to enlarge the bone defect into a regular one by open surgery, which increases the difficulty and cost of the operation, and brings severe trauma, pain and severe dysfunction to the patient. As a result, the recovery rate of the patient's wound is also slowed. However, the shape memory SF/MgO composite scaffolds can be processed into large blocks, which can be trimmed during the operation according to demand. The aim is to avoid enlarging the irregular bone defect into a regular bone defect. We anticipate that these bioactive shape memory porous scaffolds have great clinical application potential in minimally invasive clinical-size irregular bone regeneration.

“Page 3: "Calcium phosphate bone cement (CPC) is a popular injectable biomaterial owing to its well biocompatibility, osteoconductivity and its fitting ability,... Instead of "fitting ability", I suggest "ability to fill irregularly shaped defects prior to cure, although this is associated with exotherms and shrinkage" or something to that effect.”

Response: Thanks for your valuable comment. The detailed content has been revised in the manuscript with red font. “Calcium phosphate bone cement (CPC) is a popular injectable biomaterial in orthopaedic surgeries because of the good biocompatibility, osteoconductivity and the ability to fill the irregularly shaped defects, and it has been used for various micro-invasive patterns (such as vertebroplasty and arthroscopic bone repairing).”

“Shape memory polymers (SMPs), a kind of emerging intelligent biomaterials, have also brought great hope to patients with bone defects. " Additional references are needed.”

Response: Thanks for your valuable comment. The references have been added accordingly.

“However, traditional silk fibroin materials do not have shape memory effect, and the structure will be permanently damaged under external force, resulting in a tight fit with bone defects after implantation." This sentence is contradictory and does not make sense.”

Response: Thanks for your valuable comment. The detailed content has been revised in the manuscript with red font. “Traditional SF materials do not have a shape-memory effect, and their structure is permanently damaged under external force, resulting in a poor integration fit with bone defects after implantation.”

“Overall, the authors go back-and-forth between "water-responsive" and "blood-responsive". The authors should highlight water-responsive shape memory systems in the literature.”

Response: Thanks for your valuable comment. As suggested by the reviewer, we have unified the two expressions as **water-responsive** shape memory in the manuscript.

“Even with blood-responsivity, there is a potential "thermoreponsive" aspect for deployment at body temperature. [e.g., J. Shang, et al, Polym. Chem., 2019, 10, 1036-1055. M.R. Pfau, et al. J. Mater. Chem. B, 2021,9,4287-4297]”

Response: Thanks for your valuable comment. Around 60 °C, Silk fibroin displays an endothermic relaxation, usually related to an increase of the amorphous domain motion or protein-water bonding rupture, which induces an increase in the molecular motions and in the plasticity of SF materials [1-3]. We also verified the thermal response characteristics of the SF-1nMgO scaffolds. The results showed that the deformed dry SF-1nMgO scaffolds cannot recover the original shape at 37 °C [Supplementary Video 7]. Silk fibroin has a sophisticated hierarchical structure consisting of less ordered hydrophilic (amorphous domains) and crystallizable hydrophobic blocks (β -crystals domains) [4, 5]. Hydrophilic blocks provide solubility in water and are responsible for fibroin elasticity and toughness. Therefore, according to the performance characteristics of the silk fibroin, the responsiveness of the scaffolds at room temperature and body temperature is determined by less ordered hydrophilic blocks.

- [1] Hu, X., Kaplan, D., Cebe, P. Effect of water on the thermal properties of silk fibroin. *Thermochim. Acta* **461**, 137-144 (2007).
- [2] Marelli, B. N. et al. Programming function into mechanical forms by directed assembly of silk bulk materials. *Proc. Natl. Acad. Sci. U. S. A.* **114**, 451-456 (2017).
- [3] Tulachan, B. et al. Electricity from the Silk Cocoon Membrane, *Sci. Rep.* **4**, 5434 (2014).
- [4] Atkins, E. Silk's secrets. *Nature* **424**, 1010-1010 (2003).
- [5] Sahoo, J.K., Hasturk, O., Falcucci, T., Kaplan, D.L., Silk chemistry and biomedical material designs. *Nat. Rev. Chem.* DOI:10.1038/s41570-023-00486-x (2023).

Results

“Page 6 and Figure 1: Suggest putting scaffold composition notation ["SF", "SF-1nMgO", and "SF-3nMgO"] in both text of page 6 and Figure 1 to improve manuscript clarity.”

Response: Thanks for your valuable comment and advice. The detailed content has been added accordingly and in the revised manuscript with red font.

“According to the mass ratio of the nano-MgO particles to the solute mass of the SF solution at 0 wt%, 10 wt% and 30 wt%, named SF, SF-1nMgO and SF-3nMgO, respectively.”

“Page 5: "After 24 h of cryogelation, scanning electron microscopy (SEM) images confirmed that the three scaffolds had continuous porous structures with pore sizes of ~100 um (Fig. 2a, and Supplementary Fig. 2, 3)." Per page 33, no information is given in regard to specimen preparation for SEM. If this involves freeze-drying, the authors must acknowledge that this process can alter/introduce artifacts. Thus, a literal interpretation of the SEM images should be tempered.”

Response: Thanks for your valuable comment and advice. The specimen preparation for SEM has been supplemented and illustrated in the Experimental Methods section. The scaffolds do need to be lyophilized prior to SEM testing. In this process, for structurally stable scaffolds, our team found that the effect on the pore size of the scaffolds was still very limited during freeze-drying.

“The pore size and micro-morphology of SF/MgO scaffolds were examined by scanning electron microscope with energy dispersive spectrometer (SEM-EDS, S-4800, Hitachi, Japan) at an accelerating voltage of 10 kV. The cryogelled SF/MgO composite scaffolds were first frozen at -20 °C for 12 h and vacuum-dried at -50 °C for 24 h. The dry SF/MgO scaffolds were sputter-coated with gold for SEM test.”

“Water content is not reported; this will impact mechanical properties, and responsiveness to blood/water. Mechanical properties are only related to MgO content, but accompanied reduced water content should be considered.”

Response: Thanks for your valuable comment. As stated by the reviewer, the water content of the scaffolds has a very strong influence on its mechanical properties and water responsiveness. The test conditions have been supplemented and illustrated in the

Experimental Methods section. To equilibrate, the scaffold samples were fully saturated in PBS prior to mechanical testing. Therefore, there is a direct relationship between the mechanical properties of scaffolds and MgO content.

“The mechanical retention of samples was incubated at 37°C in SBF solution and removed at indicated time points (0, 14 and 28 days). On day 0, all samples were adequately hydrated in PBS for 12 h before testing. The compression test was performed at a strain rate of 30%/min. Young’s modulus of scaffolds was calculated from the initial linear strain range of the curve obtained from the stress-strain. Cyclic compression testing of the SF-1nMgO scaffolds was performed with a maximum strain of 30% for five cycles. Dynamic cyclic compression testing of the SF-1nMgO scaffolds was performed at a strain rate of -100%/min to -22% strain and 100%/min to 2% strain for 1,000 cycles.”

“Compressive moduli of the scaffolds are <0.3 MPa. The authors should comment on the limitation of this in the context of bone healing (e.g., in situation where load is applied).”

Response: Thanks for your valuable comment and advice. The scaffolds in this study are mainly applied to bone repair at non-load-bearing sites. Using its elastic, adaptive and tailorable properties, it can achieve minimally invasive treatment of irregular bone defects. As noted, the scaffolds do indeed have limitations, which do not meet the bone repair at load-bearing sites. We have therefore revised the relevant claims in the Discussion.

“There are still limitations to our study. The compression modulus of the SF/MgO composite scaffolds is between 0.1 and 0.3 MPa, which is suitable for non-load-bearing bone repair; an irregular-shaped bone defect model should be established that could fully demonstrate and reflect the performance of the SF/MgO composite scaffolds. The time spent studying animal models needs to be extended to evaluate the long-term bone repair effects of SF/MgO composite scaffolds. In future studies, a large experimental animal model should be established to further verify the concept of minimally invasive treatment of irregular bone defects and to explore the related molecular mechanism of osteogenesis.”

“Figure 4: What was the temperature of the shape recovery tests (i.e., was water and blood temperature RT or body temperature)? This should be added to the Experimental Section (e.g., specimen size, test temperature, etc).”

Response: Thanks for your valuable comment and advice. The shape recovery test of the scaffolds was performed under room temperature conditions. The experimental section has been thoroughly checked and optimized.

“The SF/MgO scaffolds were fully hydrated in PBS solution and subsequently subjected to bending, compression and torsion deformation, followed by exposure to room-temperature water to evaluate their ability to restore their original shape. Multiple scaffolds were completely compressed together and subsequently exposed to room-temperature water and blood to record the rate and extent of recovery of the scaffolds

to their original shape in different media environments. In addition, the compressed wet SF-1nMgO scaffolds were placed at a weight of 50 g. The scaffolds were then incubated in room-temperature water and the images were recorded. Furthermore, the potential of SF-1nMgO scaffolds to be trimmed during clinical surgery for personalised implantation was explored. The bulk SF-1nMgO scaffolds were prepared and then roughly and precisely trimmed according to the shape of the bone defect. Finally, the SF-1nMgO scaffolds were wetted and compressed to achieve the implantation of small pore-size irregular bone defects.”

Response to Reviewer #2:

This is an interesting manuscript. But the experimental design and data analysis need to be improved. Following questions need to be answered.

Response: Thank you very much for your affirmation of this work. We have tried our best to address these concerns and improve the clarity of this work. As suggested, the experimental design and data analysis have been comprehensively improved in this manuscript.

#1. How to ensure that the scaffolds is sterile before being implanted into the body?

Response: Thanks for your valuable comment. The detailed content has been added to the experimental method. “The scaffolds were sterilized by cobalt-60 (25kGy) radiation for 2 hours and then implanted.”

#2. In line 255, the author believes that "SF/Mg composite scaffolds had good blood-responsive shape memory effect". In my opinion, the scaffolds have good shape memory ability, but neither the Fig3-4 nor the supplementary video can prove that water or blood is the reason why the scaffolds return to its original shape. Because in the video, the scaffolds are put into a bone mold or in water under external extrusion, and in a dry environment, the scaffolds also show a great degree of shape recovery.

Please provide data on the volume change of the scaffolds in the air within 15 seconds after the water is squeezed out without interference from external forces.

Response: Thanks for your valuable comment. As the reviewer states, the scaffolds have good shape memory ability and show a great degree of shape recovery. We made a prominent statement and a critical experiment on this issue. The details are as follows:

Firstly, scaffolds in dry state will undergo permanent deformation after being subjected to external forces and do not have shape recovery ability [Fig. 1]. However, when the deformed dry scaffolds were in contact with water/blood, they would gradually restore their original shape. At the same time, we also admit that scaffolds in wet state or with a certain water content will be deformed under the action of external force, and scaffolds have a certain shape recovery ability after external force unloading. However,

it should be emphasized that the scaffolds do not have the ability to completely restore the original shape without contact with water/blood. Therefore, either dry or wet state deformed scaffolds have the ability to fully recover the original shape only when they are in contact with water or blood [Fig. S5 and Supplementary Video 6]. This is also the driving force for the scaffolds to achieve complete rebound and close contact with irregular tissues after being compressed and implanted into the body. Finally, we also verified the volume change value of the scaffolds within 30 seconds after the water is squeezed out without interference from external forces. After our multiple estimates, the volume of the scaffolds after rebound was ~80 % of the original volume.

“Fig. 4b, Supplementary Fig. 5 and Supplementary Videos 4-7 provided a robust demonstration that deformed SF/MgO scaffolds have the ability to fully recover the original shape only when in contact with water or blood.”

Fig. S5 Schematic illustration of water-responsive shape-memory effect SF-1nMgO scaffolds. The SF-1nMgO scaffolds in the dry state will be permanently deformed after being subjected to external force, and has no shape recovery ability. However, the deformed dry scaffolds were in contact with water/blood, they would gradually restore their original shape.

#3. Please explain how water or blood speeds up the scaffold's rebound?

Response: Thanks for your valuable comment. There are three main reasons why water or blood is able to accelerate scaffolds rebound:

1. The hydrophilic/hydrophobicity of the material itself has a direct impact on the water-/blood- responsive capacity of the scaffolds.

(i) Silk fibroin has a sophisticated hierarchical structure consisting of less ordered hydrophilic (amorphous domains) and crystallizable hydrophobic blocks (β -crystals domains) [4, 5]. Hydrophilic blocks provide solubility in water and are responsible for fibroin elasticity and toughness, while hydrophobic blocks form intermolecular β -sheet structures leading to the insolubility and high strength of fibroin.

2. The scaffolds have excellent water-/blood-responsive shape-memory effect, which is inseparable from excellent mechanical stability and structural retention.

(i) The mechanical properties of silk scaffolds are dominated by its cross-linked configuration, and SF amorphous domains firstly support the external forces [6]. The scaffolds in this study are multiple cross-linking network structural materials integrating covalent cross-linking, physical cross-linking, and electrostatic interactions. After the scaffolds are subjected to compression, the SF amorphous domains will be deformed but the structure will not be damaged. In this process, internal stress will be generated in the scaffolds to restore the original state [7]. This reason is also the basis and prerequisite for scaffolds to have water or blood responsiveness.

(ii) β -crystals domain arrangements, size, and orientation also influence the mechanical properties. β -crystals confined to a few nanometers can achieve higher strength, stiffness, and toughness than larger β -crystallites. The reported dimensions of β -crystallites for optimal mechanical properties are between 1 and 4 nm [8]. It is worth noting that our previous research work has demonstrated that β -crystals of SF scaffolds prepared by our optimized cryogelation technology are confined to 1-2 nanometers [9]. This is another reason why the scaffolds have blood-responsive shape-memory effect. The relevant explanations and analyses have been added and in the revised manuscript with red font.

“The main explanation for this phenomenon falls into three categories: (i) SF has a sophisticated hierarchical structure consisting of less ordered hydrophilic (amorphous domains) and crystallisable hydrophobic blocks (β -crystals domains). Hydrophilic blocks provide solubility in water and are responsible for fibroin elasticity and toughness; (ii) the SF/MgO composite scaffolds are subjected to compression, and the SF amorphous domains deform first, but the structure remains intact. During this process, internal stress is generated in the scaffolds to restore the original state; and (iii) domain arrangements, size and orientation of the β -crystals also influence the mechanical properties. β -crystals of SF scaffolds prepared by our optimised cryogelation technology are confined to 1-2 nanometers, which can achieve higher strength, stiffness and toughness.”

[4] Atkins, E. Silk's secrets. *Nature* **424**, 1010-1010 (2003).

[5] Sahoo, J.K., Hasturk, O., Falcucci, T., Kaplan, D.L., Silk chemistry and biomedical material designs. *Nat. Rev. Chem.* DOI:10.1038/s41570-023-00486-x (2023).

[6] Reizabal, A., Costa, C.M., Pérez-Alvarez, L., Vilas-Vilela, J.L. & Lanceros-Méndez, S. Silk Fibroin as Sustainable Advanced Material: Material Properties and Characteristics, Processing, and Applications. *Adv. Funct. Mater.* **33**, 2023.

[7] Ak, F., Oztoprak, Z., Karakutuk, I., Okay, O. Macroporous Silk Fibroin Cryogels. *Biomacromolecules* **14**, 719-727 (2013).

[8] Koh, L.D. et al. Structures, mechanical properties and applications of silk fibroin materials. *Prog. Polym. Sci.* **46**, 86-110 (2015).

[9] Mao, Z.N. et al. Controlled cryogelation and catalytic cross-linking yields highly elastic and robust silk fibroin scaffolds. *ACS Biomater. Sci. Eng.* **6**, 4512-4522 (2020).

#4. The author believes that the scaffolds can be used for Irregular Bone Regeneration, but only a critical-sized skull defects model was designed in the experiment, and an irregular-shaped bone

defect model was not established on this basis.

Response: Thanks for your valuable comment. We regret that the irregular-shaped bone defect model was not established in this work. In fact, the establishment of irregular bone defect model is still in the development stage, and there is no unified standard. And more scholars are needed to participate in the discussion of the establishment of irregular bone defect model. Nevertheless, the previous materials research on the repair and regeneration of irregularly shaped bone defects were still mainly based on critical-sized skull defects model [10-13]. In the future, we will revise the animal defect model for the evaluation of the effects of irregular-shaped bone repair and regeneration.

[10] Liu, Q. et al. Highly Malleable Personalized Prostheses with Hierarchical Microstructure Boost the Long-Term Osteointegration in Irregular Craniofacial Reconstruction. *Adv. Funct. Mater.* DOI:10.1002/adfm.202304308 (2023).

[11] Li, Y.M. et al. Injectable Biomimetic Hydrogel Guided Functional Bone Regeneration by Adapting Material Degradation to Tissue Healing. *Adv. Funct. Mater.* DOI:10.1002/adfm.202213047 (2023).

[12] Zou, Y.P. et al. Precipitation-Based Silk Fibroin Fast Gelling, Highly Adhesive, and Magnetic Nanocomposite Hydrogel for Repair of Irregular Bone Defects. *Adv. Funct. Mater.* **33**, (2023).

[13] Lu, G.G. et al. An instantly fixable and self-adaptive scaffold for skull regeneration by autologous stem cell recruitment and angiogenesis. *Nat. Commun.* **13**, (2022).

Response to Reviewer #3:

The authors reported silk fibroin/magnesium composite scaffolds with improved mechanical properties and shape-memory combining chemical and physical crosslinking for irregular bone regeneration. This scaffold could promote cell proliferation, adhesion, migration of osteoblasts, and osteogenic differentiation of BMSCs in vitro and could promote in situ bone regeneration of cranial defect, which are interesting. However, the statement "excellent mechanical performance" is not convinced and whether the degradability of such composite scaffold is suitable for bone regeneration is not clearly supported, I think. Previous studies have proved that magnesium could also regulate the cell functions including proliferation, adhesion, and migration, as well as promote neovascularization. I cannot see any milestones achieved to meet the standard of Nature Communications; I believe. Some comments are as follows:

Response: Thank you very much for your interest and the nice comment on this work. The statement "excellent mechanical performance" has been revised to "excellent mechanical stability". According to the reviewer's comments, whether the degradation rate of the scaffolds is suitable for bone regeneration, we have therefore revised the relevant claims in the discussion. As the reviewer's comments, previous studies have proved the biological functions of Mg²⁺. Our group have also made great efforts on the role of Mg metal/Mg²⁺ in promoting osteogenesis, and revealed the mechanism of Mg²⁺ in promoting bone regeneration [14-16]. Meanwhile, our previous study has in-depth research on the relationship between the molecular structure of silk fibroin and its

physicochemical and biological properties [17, 18]. At the same time, the regeneration of critical-sized bone defects, especially for the ones with irregular shapes, remains a clinical challenge. Therefore, based on our previous extensive studies, this work creatively constructed a water-responsive shape memory effect silk fibroin/ magnesium (SF/MgO) composite scaffold, which was able to easily trimmed, compacted and implanted into the body, and fully recovered original shape in contact with water or blood, and it could provide mechanical support for tight-contact with surrounding tissues. The addition of MgO particles in SF scaffolds obviously enhanced the biological activity; on the other hand, it brought more possibilities for the regulation of the degradation rate of scaffolds for bone repair. The SF/MgO composite scaffold basically meets the requirements of minimally invasive implantation of irregular bone defects.

“In vivo cranial bone repair generally took over 8 weeks, indicating that the degradation duration of scaffolds should meet the requirement of the healing time. At 8 weeks after SF-1nMgO scaffold implantation, the histological staining results showed partial degradation of the SF-1nMgO scaffolds, accompanied by in situ regeneration of new bone tissue inside the scaffolds. Therefore, a prolonged implantation time of the SF-1nMgO scaffolds meant the remaining MgO inside the degrading scaffolds would gradually hydrolyse and release Mg^{2+} to promote bone repair. By contrast, SF scaffolds were largely degraded after 8 weeks of implantation, but very limited new bone formation was seen. Therefore, while the addition of MgO particles to SF scaffolds enhanced the biological activity, it also brought more possibilities for regulating the degradation rate of scaffolds for bone repair.”

- [14] Shen, J. et al. Stepwise 3D-spatio-temporal magnesium cationic niche: Nanocomposite scaffold mediated microenvironment for modulating intramembranous ossification. *Bioact. Mater.* **6**, 503-519 (2021).
- [15] Wong, H.M. et al. Low-modulus Mg/PCL hybrid bone substitute for osteoporotic fracture fixation. *Biomaterials* **34**, 7016-7032 (2013).
- [16] Zhang, Y.F. et al. Implant-derived magnesium induces local neuronal production of CGRP to improve bone-fracture healing in rats. *Nat. Med.* **22**, 1160-1169 (2016).
- [17] Mao, Z.N. et al. Controlled Cryogelation and Catalytic Cross-Linking Yields Highly Elastic and Robust Silk Fibroin Scaffolds. *ACS Biomater. Sci. Eng.* **6**, 4512-4522 (2020).
- [18] Mao, Z.N. et al. The relationship between crosslinking structure and silk fibroin scaffold performance for soft tissue engineering. *Int. J. Biol. Macromol.* **182**, 1268-1277 (2021).

1. The mechanical properties of the SF/Mg composite scaffolds were shown in Fig. 3, around few hundred kPa, which are not high for bone implants and bone tissue. Please comment it.

Response: Thanks for your valuable comment. The scaffolds in this study are mainly applied to bone repair at non-load-bearing sites. Using the elastic, adaptive and tailorable properties of SF/MgO scaffolds, it can achieve minimally invasive treatment of irregular bone defects. Ideally, biomaterials with excellent osteoinductivity, suitable porosity, and biodegradability are preferred for non-load-bearing bone repair. More

importantly, the degradation rate of materials should match the bone regeneration rate, providing sufficient scaffold-derived support for cell growth and differentiation, meanwhile degrading to provide space for new bone growth [11, 19]. The detailed content has been added accordingly and in the revised manuscript with red font. “The elastic modulus was between 0.1 and 0.3 MPa, which was suitable for non-load-bearing bone repair”.

[11] Li, Y.M. et al. Injectable Biomimetic Hydrogel Guided Functional Bone Regeneration by Adapting Material Degradation to Tissue Healing. *Adv. Funct. Mater.* DOI:10.1002/adfm.202213047 (2023).

[19] Lee, S.S. et al. Sequential growth factor releasing double cryogel system for enhanced bone regeneration. *Biomaterials* **257**, (2020).

2. Is it strange that "The addition of Mg particles reduced the degradation rate of SF scaffolds and "the degradation rate of the three scaffolds followed the order: SF-3nMgO > SF-1nMgO > SF (Fig. 2f)" and "However, the degradation rate of the three groups is not different. "on lines 195-200? Whether the degradability of such composite scaffold is suitable for bone regeneration is not clearly supported.

Response: Thanks for your valuable comment. We apologize for the expression of "However, the degradation rate of the three groups is not different.". What we intend to express is that there is no significant difference in the conformational content of the three groups of materials. It must be admitted that the matching of the degradation rate of the materials with the regeneration rate of the new bone tissue is the core point of the design of the new materials. The *in vitro* degradation speed test of materials cannot provide a scientific criterion for the degradation trend *in vivo*, but it can be used to compare the degradation trend of materials under different parameters. According to the reviewer's comments, whether the degradation rate of the scaffolds is suitable for bone regeneration, we carry out a detailed and in-depth analysis in the discussion section.

“*In vivo* cranial bone regeneration generally took over 8 weeks, indicating degradation duration of scaffolds should meet the requirement of the tissue regenerate time. At 8 weeks after SF-1nMgO scaffolds implantation, the histological staining results showed partial degradation of the SF-1nMgO scaffolds, accompanied by *in situ* regeneration of new bone tissue inside the scaffolds. Therefore, if the implantation time of the SF-1nMgO scaffolds is prolonged; with the degradation of the remaining scaffolds, the remaining MgO inside the scaffolds will gradually hydrolyze and release Mg^{2+} to promote bone regeneration and repair. In contrast, SF scaffolds were largely degraded after 8 weeks of implantation, but there was limited new bone regeneration. Therefore, on the one hand, the addition of MgO particles in SF scaffolds enhances the biological activity of scaffolds; on the other hand, it brings more possibilities for the regulation of the degradation rate of scaffolds for bone repair.”

3. How about the data of the in vivo bone regeneration after longer time such as 12-weeks, because bone regeneration needs longer time.

Response: Thanks for your valuable comment and suggestion. We regret that the animal defect model was performed for 8 weeks. At 8 weeks after implantation, the micro-CT, histological and immunological staining analysis had showed that the SF/MgO group was significantly better than the blank control group and the SF group. Meanwhile, the data 8-week of SF/MgO group in this study was higher than the 8-week data and even close to the 12-week data of other studies through the horizontal comparison of bone volume fraction [19-21]. Nevertheless, previous studies on bone defect repair were conducted for 4 or 8 weeks [21-25]. At 8 weeks after scaffolds implantation, the histological staining results showed that partial degradation of the scaffolds occurred, accompanied by in situ regeneration of new bone tissue inside the scaffolds. It is well known that silk fibroin is an enzyme-responsive, fully degradable material. Therefore, if the implantation time of the scaffolds is prolonged; with the degradation of the remaining scaffolds, the remaining MgO inside the scaffolds will gradually hydrolyze and release Mg²⁺ to promote bone regeneration and repair. Therefore, we hypothesized that the bone repair effect of the scaffolds could be further improved by prolonging the implantation time. In the future, we will extend the animal model time for the evaluation of the effects of bone repair and regeneration.

- [19] Lee, S.S. et al. Sequential growth factor releasing double cryogel system for enhanced bone regeneration. *Biomaterials* **257**, (2020).
- [20] Zhang, Y.C. et al. 3D-printed NIR-responsive shape memory polyurethane/magnesium scaffolds with tight-contact for robust bone regeneration. *Bioact. Mater.* **16**, 218-231 (2022).
- [21] Qiao, W. et al. TRPM7 kinase-mediated immunomodulation in macrophage plays a central role in magnesium ion-induced bone regeneration. *Nat. Commun.* **12**, (2021).
- [22] Zhao, Z.Y. et al. Capturing Magnesium Ions Microfluidic Hydrogel Microspheres for Promoting Cancellous Bone Regeneration. *ACS Nano* **15**, 13041-13054 (2021).
- [23] Zhang, X.D. et al. Bioinspired Mild Photothermal Effect-Reinforced Multifunctional Fiber Scaffolds Promote Bone Regeneration. *ACS Nano* **17**, 6466-6479 (2023).
- [24] Liu, W.W. et al. In situ activation of flexible magnetoelectric membrane enhances bone defect repair. *Nat. Commun.* **14**, (2023).
- [25] Chen, S. et al. Local H₂ release remodels senescence microenvironment for improved repair of injured bone. *Nat. Commun.* **14**, (2023).

4. Residual crosslinker EGDE was not removed? How about the toxicity?

Response: Thanks for your valuable comment. In our previous work, attenuated total reflectance-fourier transform infrared spectroscopy (ATR-FTIR) results have demonstrated that residual EGDE can be completely removed from the scaffolds [9]. At the same time, the *in vitro* cytocompatibility of the SF scaffolds was assessed with bone marrow mesenchymal stem cells (BMSCs). The results showed that the cells had a good proliferation trend and cell morphology with the increase of culture time [9, 26].

- [9] Mao, Z.N. et al. Controlled cryogelation and catalytic cross-linking yields highly elastic and robust silk fibroin scaffolds. *ACS Biomater. Sci. Eng.* **6**, 4512-4522 (2020).
- [26] Mao, Z.N. et al. A Cell-Free Silk Fibroin Biomaterial Strategy Promotes In Situ Cartilage Regeneration Via Programmed Releases of Bioactive Molecules. *Adv. Healthc. Mater.* **12**, e2201588 (2023).

5. On Line 149 "EGDE"? Line 192, what are "SF-1nMgO, and SF-3nMgO"? "Silk Fibroin/Magnesium" on lines 275-276, why are the uppercase letters used here? Some errors such as line 390 "8 weeks after surgery, The 3D reconstruction image as shown in Fig. 9b.

Response: Thanks for your valuable comment.

In the revised manuscript, we have given specific explanations for "EGDE", "SF-1nMgO", and SF-3nMgO".

“ethylene glycol diglycidyl ether (EGDE)”

“According to the mass ratio of the nano-MgO particles to the solute mass of the SF solution at 0 wt%, 10 wt% and 30 wt%, named SF, SF-1nMgO and SF-3nMgO, respectively.”

"Silk Fibroin/Magnesium" on lines 275-276 has been revised to “silk fibroin/magnesium”.

“line 390 "8 weeks after surgery, The 3D reconstruction image as shown in Fig. 9b.” has been revised to “First, micro-CT analysis was used to assess bone regeneration at 4 and 8 weeks after surgery. Fig. 9b shows the 3D reconstruction images.”

The manuscript has been thoroughly checked and optimized, including grammar, experimental design, data analysis and some detail errors.

Reviewers' Comments:

Reviewer #1:

Remarks to the Author:

Reviewer comments are adequately addressed.

Reviewer #3:

Remarks to the Author:

OK !

Reviewer #4:

Remarks to the Author:

[Note from the Editor: Reviewer #4 was invited to review the response given to the original Reviewer #2.]

In this manuscript, Zheng et al. have successfully developed a novel class of water-responsive and shape-memory porous SF/MgO scaffold via cryogelation. As evidenced by systematic in vitro and in vivo experimental results, the as-constructed scaffolds exhibited excellent pro-osteogenic and pro-angiogenic properties. The authors have made thorough revisions to address most concerns of all reviewers, including the improvement in the experimental design and data analysis according to the comments of reviewer #2. I think that the revised manuscript meets the high publication standards of Nature Communications.

Response to Reviewer #1:

Reviewer comments are adequately addressed.

Response: Thank you very much for your affirmation of the response.

Response to Reviewer #3:

OK!

Response: Thank you very much for your affirmation of the response.

Response to Reviewer #4:

In this manuscript, Zheng et al. have successfully developed a novel class of water-responsive and shape-memory porous SF/MgO scaffold via cryogelation. As evidenced by systematic in vitro and in vivo experimental results, the as-constructed scaffolds exhibited excellent pro-osteogenic and pro-angiogenic properties. The authors have made thorough revisions to address most concerns of all reviewers, including the improvement in the experimental design and data analysis according to the comments of reviewer #2. I think that the revised manuscript meets the high publication standards of Nature Communications.

Response: Thank you very much for your affirmation of this work.